# Global reward state affects learning and activity in raphe nucleus and anterior insula in monkeys

Marco K. Wittmann [1✉], Elsa Fouragnan [1,2], Davide Folloni [1], Miriam C. Klein-Flügge [1], Bolton K. H. Chau [3], Mehdi Khamassi [4] & Matthew F. S. Rushworth[1,5]

People and other animals learn the values of choices by observing the contingencies between them and their outcomes. However, decisions are not guided by choice-linked reward associations alone; macaques also maintain a memory of the general, average reward rate – the global reward state – in an environment. Remarkably, global reward state affects the way that each choice outcome is valued and influences future decisions so that the impact of both choice success and failure is different in rich and poor environments. Successful choices are more likely to be repeated but this is especially the case in rich environments. Unsuccessful choices are more likely to be abandoned but this is especially likely in poor environments. Functional magnetic resonance imaging (fMRI) revealed two distinct patterns of activity, one in anterior insula and one in the dorsal raphe nucleus, that track global reward state as well as specific outcome events.

[1] Wellcome Centre for Integrative Neuroimaging (WIN), Department of Experimental Psychology, University of Oxford, Oxford, UK. [2] School of Psychology, University of Plymouth, Plymouth, UK. [3] Department of Rehabilitation Sciences, The Hong Kong Polytechnic University, Hong Kong, Hong Kong. [4] Sorbonne Université, CNRS, Institute of Intelligent Systems and Robotics, F-75005 Paris, France. [5] Wellcome Centre for Integrative Neuroimaging (WIN), Centre for Functional MRI of the Brain (MRI), Nuffield Department of Clinical Neurosciences, John Radcliffe Hospital, University of Oxford, Oxford, UK. ✉email: marco.k.wittmann@gmail.com

Humans and animals make a multitude of choices every day. Ideally the choice taken should be the one with the highest value – the one most likely to yield positive results. It is widely held that choice value estimates reflect past experience of the consequences of making the same choices; if a choice has led to a positive outcome in the past then it has a higher value than one lacking such a consequence[1–3]. A choice outcome that is better than previously expected generates a positive prediction error (PE) and, as a consequence, increases the choice's value. Conversely, negative PEs decrease a choice's value estimate. Therefore, implicit in such models is the notion that the experience of choice-reward conjunctions determines choice valuations.

While substantial evidence suggests that this is indeed the case and that such a process can be captured in reinforcement learning (RL) models[2–5] there is evidence that other aspects of choice experience are also important determinants of whether they will be taken again. For example, there is a tendency for both animals and humans simply to repeat previous choices[6,7]; taking the choice on one occasion makes it more likely that it will be taken again on another. In addition, there is evidence for the existence of non-conjunctive effects – the forging of inappropriate links between an unrelated choice and reward[8–14].

Here we first report analyses of decision-making behavior in four macaques. We show that in addition to conjunctive choice-reward associations, unlinked memories of both choice and reward persist over several trials and influence current and future choices the monkeys take. Rewards related to the latter effect – an influence of reward experience that is unlinked to any particular choice – we term global reward state (GRS). It has a striking impact on behavior: Animals stayed increasingly with rewarded choices if those were encountered in high GRS, while they abandoned poor choices particularly when encountered in low GRS. This meant that a low GRS drove animals to explore alternative choices. Drawing on previous models[15–18], we constructed a new RL model that captures both conjunctive choice-reward associations and GRS effects to explain the monkeys' behavior.

Finally, we applied the RL model to neural activity recorded with fMRI. Activity in two areas, the anterior agranular insular cortex (Ia) adjacent to posterior orbitofrontal cortex (OFC) and the dorsal raphe nucleus (DRN) reliably reflected specific instances of rewards and non-rewards but also the GRS. The temporal pattern of activity in Ia suggested Ia gradually integrates new rewards into a longer-term global estimate of value. By contrast, the DRN pattern suggested a role in regulating the impact that any negative event will have as a function of GRS.

## Results

### Task structure

We analyzed behavioral data from four macaque monkeys in three experiments using a binary-choice probabilistic bandit task (overall 65 sessions of 200 trials each). In addition, we also analyzed neural data from one of the three experiments (25 sessions)[19,20]. Choice options were allocated pseudorandomly to the right and left side of the screen and monkeys responded towards congruently located right or left sensors. Choice outcomes were either a drop of juice or nothing (Fig. 1a). The rewards were delivered probabilistically and the probabilities of two of the options reversed towards the middle of a session and monkeys' choice frequencies followed this reversal (Fig. 1b).

Importantly, each session contained three new choice stimuli, but only two of them were choosable on each trial (Fig. 1c). This was crucial in several respects. First, it enabled dissocation of choice repetition effects linked to either a choice's location or target stimulus. Second, selective choice presentation and the probabilistic nature of the task ensured that effects of the GRS were dissociable from conjunctive choice-reward effects and that the GRS would fluctuate continuously over the course of a session (Fig. 1d). Finally, it ensured that the GRS would be dissociable from reversals in the relative rates of reward associated with different options (compare Fig. 1b, d).

### Behavioral effects of unlinked choice and reward memories

To discover potential effects of unlinked memory traces of choice and reward on decision making, we conceptualized monkeys' binary choices as stay/switch decisions to either continue a current course of action or switch to an alternative[21–23]. We constructed a detailed logistic general linear model (GLM) for which we concatenated data per monkey per experiment, pooling data over all three experiments ($n = 12$ monkey data sets; 4 macaques). For every trial $t$, we identified the chosen stimulus C and examined whether the animals would choose C again the next time C was offered (Fig. 1e). We tested whether the stay/switch decision was predicted, first, by the conjunctive choice-reward history of C (CxR-history), second, by the (reward-unlinked) choice history of C (C-history) and, third, by the (choice-unlinked) reward history (R-history; note that the regressor construction is explained in detail in Supplementary Fig. 1a).

We found that all three sets of regressors, CxR-history, C-history, and R-history, significantly promoted stay behavior (Fig. 1f). Firstly, macaques stayed with options if choosing them had been rewarded recently; reward on trial $t$ increased the likelihood of repeating the choice made on that trial (one-sample $t$-test; CxR-history($t$): $t_{11} = 8.883$; $p < 0.001$). Secondly, macaques repeated choices they had made most recently irrespective of reward (C-history($t-1$): $t_{11} = 6.496$; $p < 0.001$). The strength of these effects decreased with time (main effect of recency in 5 [recency] × 2 [history type] ANOVA: $F(4,44) = 20.09$; $p < 0.001$). Finally, and most intriguingly, R-history also had a significantly positive effect on choice (R-history: $t_{11} = 4.711$; $p < 0.001$). That means that irrespective of directly reinforced choices and choice repetitions, animals repeated choices that they had previously made at times of a high GRS. In turn, they switch away from a choice - even if that specific choice has been rewarded lately – more often if it is encountered when in a low GRS. Note that this analysis also controls for the history of the upcoming alternative option (Supplementary Fig. 1b), is stable when varying the length of the reward history considered (Supplementary Fig. 1c) and even holds when analyizing monkeys individually (Supplementary Fig. 2).

These results suggest that the GRS alters the animals' response to rewards received for a current choice. To investigate this directly, we regressed all effects of the previous GLM (Fig. 1f), except those of CxR-history($t$) and R-history, out of the choice data. We then examined the residual choice probabilities (as provided by Matlab's glmfit function). We binned them, first, by the current outcome, and second, by the binarized GRS (median split of R-history; Fig. 1g). Animals were more likely to stay after a win and more likely to switch after a loss. However, in addition, staying after a win was more likely when R-history was high compared to when it was low and vice versa switching away from a loss was more likely in low reward environments ($2 \times 2$ ANOVA; interaction: $F_{1,11} = 31.68$; $p < 0.001$). This indicates, remarkably, that the monkeys not only evaluate an option as a function of the outcomes contingent on its choice, but that they do so more strongly when those outcomes are consistent with the overall reward context: In a high GRS, negative feedback has a relatively weaker impact on an options' value than it does in a low GRS; switching is less likely to occur after non-reward in a high as opposed to low GRS.

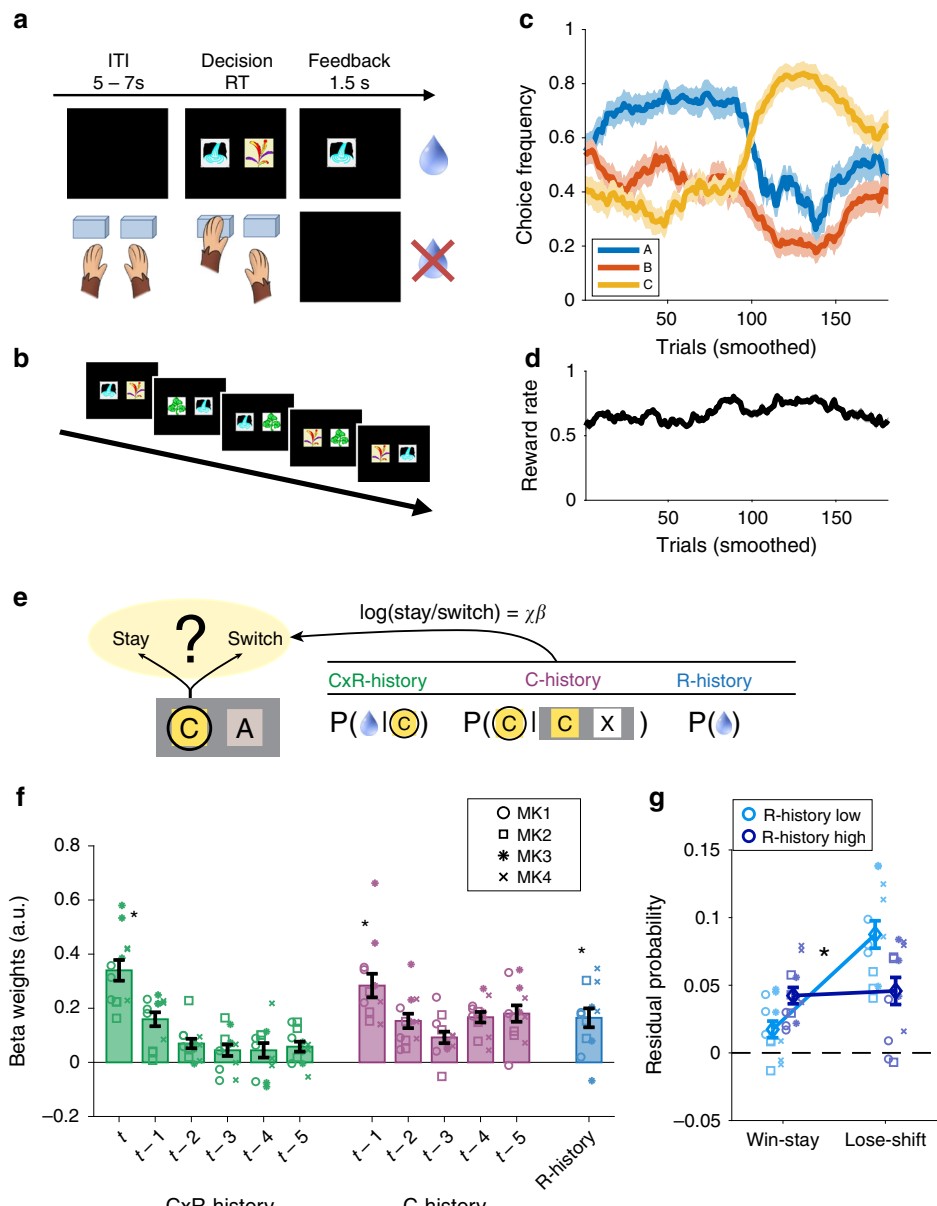

**Fig. 1 Task design and behavior. a** Monkeys performed a probabilistic binary-choice task for juice reward. **b** One session comprised three options indicated by different stimuli, only two of which were choosable each trial (experimenter determined, pseudo-random). **c** Smoothed choice frequency (moving window of 20 trials) followed the reversal in reward probabilities of two of the three options (blue and yellow). **d** The smoothed overall average reward rate (moving window of 20 trials) fluctuated continuously over the course of the session. **e** Illustration of GLM analysis. For every trial we predicted whether the chosen option (C in this example) would be repeated on the next occasion it was presented using three sets of regressors. The first set reflected the reward probability of C if it was chosen (CxR-history), the second set reflected the tendency to choose C in the past if it was one of the offered options (C-history), and the third reflected the global reward state unrelated to a specific choice (R-history). **f** Significantly positive effects of CxR-history, C-history, and R-history on the decision to stay with an option. **g** Residual probability of making a stay or switch choice. Stay/switch behavior was consistent with the outcome (win/lose) on trial t, but the effect was facilitated if the outcome was in accordance with global reward state: Switching away from an option was more likely if R-history was low, while staying with a rewarded option was more likely in high-reward contexts. (in panels **b**, **d**, data related to the MRI experiment is shown as mean values ± SEM across sessions; $n = 25$; panels **f**, **g** concatenate sessions per monkey per experiment resulting in three data points per individual; data are presented as mean values ±SEM across monkey data sets; $n = 12$; asterisks indicate $p < 0.001$; we used one-sample two-sided $t$-tests against zero and analyses of variance). Source data are provided as Source Data file. Symbols indicate monkey identity in panels **f**, **g**; MK abbreviates monkey. (Panels **a**, **c** are adapted from Chau et al.[19]).

**RL model incorporating choice memory and global reward state.** After providing evidence for the unlinked influence of choice memories and GRS on decision making, we went on to formalize a possible computational mechanism for the observed effects in an RL framework. We used hierarchical model fitting[24–26] over all three data sets collapsed.

In addition to a standard RL architecture, we added unlinked choice and reward memory traces in the model. A choice-location trace (CL-trace) and choice-stimulus traces (CS-traces) captured recency-weighted averages of past choice locations and past choice stimuli, respectively[4,27] (Fig. 2a, b). Thus CL-trace and CS-trace capture two aspects of choices – their locations and their

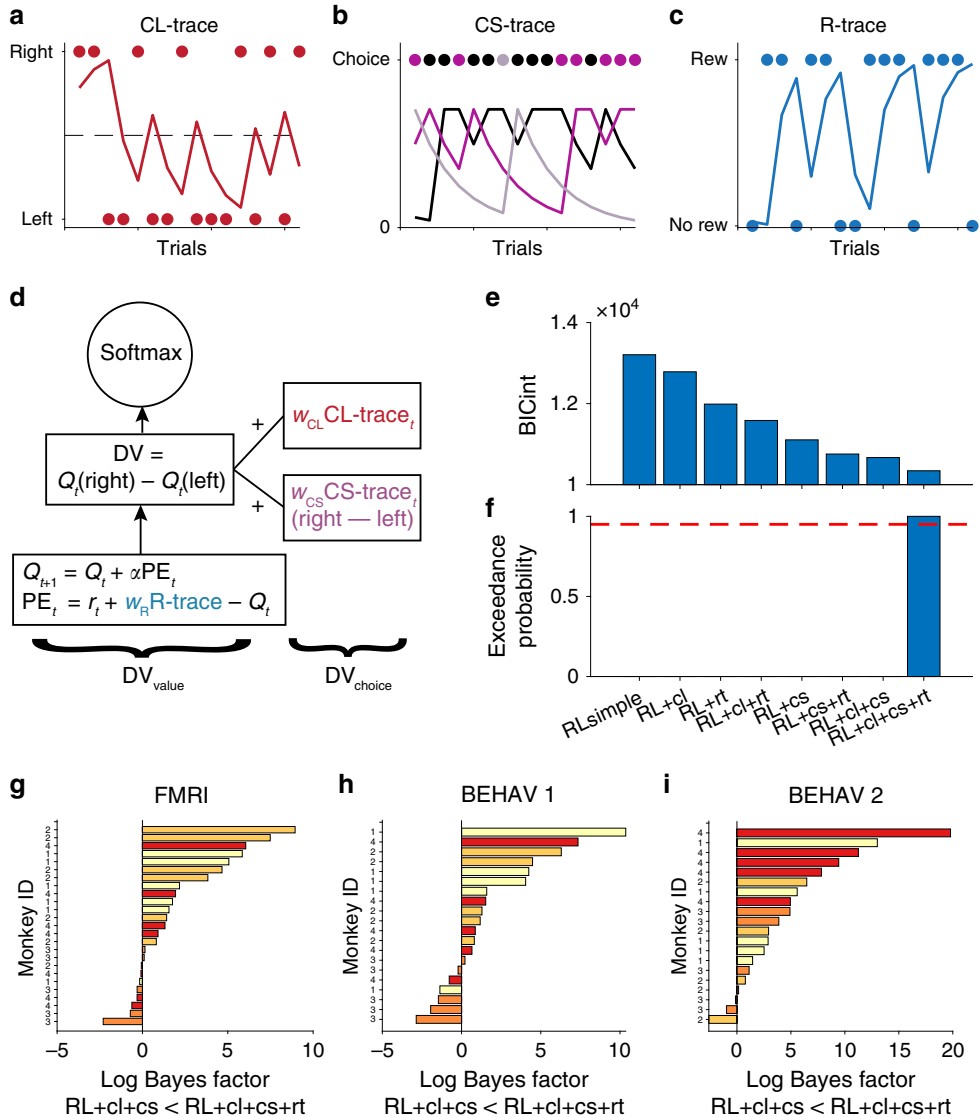

**Fig. 2 RL modeling of choice and reward traces.** Illustration of choice-location trace (CL-trace, **a**), choice stimulus traces (CS-trace, different colors indicate different choice stimuli, **b**) and reward trace (R-trace, **c**). **d** Model architecture. Black colors indicate components used by all models. The other colors refer to choice and reward traces as illustrated in panels **a–c**. Note that CL-trace and CS-trace effects are added directly to the DV of the model, whereas R-trace affects the prediction error calculation. **e** Nested model comparison using BICint (lower values indicate better fit). The full model including R-trace explains the data best. x-labels are the same for panels **e** and **f**. **f** Exceedance probability also favors the full model among our set of candidate models. The dashed red line indicates an exceedance probability of 0.95. **g–i** Experiment specific comparisons of Bayes factors between full model and second-best model. Bars are individual sessions and direction of the bars (left vs right) indicate better model fit in favor of the full (going to the right) compared to the second-best model (going to the left). Colors indicate individual monkeys. (*n* = 65 sessions). Source data are provided as Source Data file.

identities. A reward trace (R-trace) captured the recency-weighted GRS. R-trace corresponded to the disembodied average of past rewards, irrespective of the choice stimuli or choice locations linked to it (Fig. 2c).

In the simplest RL implementation (RLsimple), the decision variable (DV) was calculated as the difference in Q-value between the left and right option, (plus an added side bias, see Methods; Fig. 2d). We extended the RLsimple model by incorporating also the weighted choice trace differences in the DV. In the final model, the weights associated with both CL and CS traces were positive indicating choice repetition on both levels consistent with the behavioral analyses (Fig. 1f; Supplementary Figs. 2 and 3). In contrast to choice traces, R-trace is unlinked to specific options. For this reason, we added R-trace to the PE calculation scaled by a weight parameter $w_R$. This meant that the PE was no longer

conceived as simply the difference of expectation and outcome. Instead the PEs could be enhanced or diminished by low or high levels of global rewards. The directionality of the R-trace effect depended on $w_R$ which was empirically fitted and allowed to range between −1 and 1. Such use of R-trace is inspired by models of average reward rate learning[15,28].

In our nested model comparison, we considered models with every possible permutation of the components as well as the full model (Fig. 2e, f). We calculated the integrated Bayesian information criterion (BICint)[25] (Fig. 2e). We observed a steady decrease of BICint (indicating better model fit) when including memory traces. The full model (RL + cl + cs + rt) won the model comparison. As an additional validation, we calculated the exceedance probability, i.e., the posterior probability that a model is the most likely model used by the population among a given set

of models, and again found that the full model was the best one (exceedance probability = 1.0; Fig. 2f)[29]. These results are stable even when considering the experiments separately or the tested monkeys individually (Supplementary Fig. 3). Moreover, we used such experiment-wise fitted models to specifically compare the full model with the second best one (RL + cl + cs) via log Bayes factors (Methods; Fig. 2g–i). Over all data sets, the majority of sessions were in favor of the best over the second-best model (47 out of 65; binomial test against 0.5; $p < 0.001$).

**RL learning dynamics induced by the global reward state**. Animals learn the value of choices based on their reward consequences. Our behavioral (Fig. 1f, g) and modeling results (Fig. 2e, f) suggest that in our data, in addition, the GRS affects animals' learning of choice values. In our model, the direction of the influence of GRS on PE calculations critically depends on $w_R$. Note that supplementary model comparison indicates that our model also fits better than models where the effect of R-trace is not mediated by the PEs (Supplementary Fig. 4j, k). In principle, the model allows GRS to have negative, contrasting effects ($w_R <$ 0), no effect at all ($w_R = 0$), or positive, facilitating effects ($w_R >$ 0). We therefore examined the fitted parameter values of $w_R$, i.e the final group–level point-estimates of $\mu$ and $\sigma$ associated with $w_R$ (Methods). We found that the posterior density of the fitted $w_R$ was overwhelmingly positive for all three experiments (Fig. 3a, b).

Such positive $w_R$ lead both positive and negative PEs to be shifted in a positive direction when R-trace is high compared to when R-trace is low (Fig. 3c; sessions are fitted to the mean of the posterior). In theory it could even turn negative PEs positive when R-trace is very high. The implications of this become more transparent when reframing the PEs as absolute PEs – reflecting the degree to which outcomes suggest a need for re-evaluation regardless of direction (or surprise[24]). Then it becomes clear that positive value updates become stronger in a high GRS and negative value updates become stronger in a low GRS (Fig. 3d; median split of R-trace; 2 × 2 ANOVA interaction; $F_{1,64} = 849$; $p < 0.001$). This means that the model effectively implements asymmetric learning rates for positive and negative outcomes that change dynamically as a function of GRS. We further demonstrate this by estimating the effective learning rates from our model and indeed find a significant interaction depending on outcome type and GRS being low or high (Fig. 3e; $F_{1,64} = 18.357$; $p < 0.001$). This maps directly onto the pattern of stay-switch choices observed in the initial behavioral analysis, showing that a high GRS promotes win-stay and a low GRS promotes lose-switch behavior (Fig. 1f, g). It also explains why our model fits better than models assuming different learning rates for positive and negative outcomes (AsyAlpha) or models assuming dynamic learning rates based on the degree of surprise encountered over time (dynAlpha; Fig. 3f, g; Supplementary Fig. 4j, k). The reason is that reference to the GRS is critical to explain the dynamics of value updates in our data.

Our RL model allows rewards not only to influence concurrent choices, but also choices that occur later in time and are logically unrelated (via R-trace). For this reason, we used model simulations to examine whether our model also predicts some previously reported cases of credit misassignment[8,11,13], in particular when past rewards are mistakenly credited to subsequent choices[11,13]. We used previously reported GLM designs[11,13]. These simulations show that repeating a choice made at time point $t$ is more likely if reward is received at $t$, but strikingly also if reward is received before at $t-1$; however only if $w_R$ positive. Negative $w_R$ on the other hand lead to contrasting effects of reward (Fig. 4). This suggests that indeed a GRS

informed learning rule such as in our model can cause reward to spread forward to choices that are made after such reward is delivered.

Finally, rather than biasing decision making exclusively, the GRS may influence reaction times (RTs) as well[28,30]. We predicted the trialwise negative logarithm of RTs using linear regression (Fig. 5). Q-values of the chosen and unchosen options had no significant influence (both $t_{64} < 1.27$; both $p > 0.2$), and neither did their difference ($t_{64} = 1.57$; $p > 0.13$). Instead, animals responded quicker when stimuli were offered that had been picked more recently: when the chosen CS-trace ($t_{64} = 6.63$; $p < 0.001$) and or the unchosen CS-trace was high ($t_{64} = 3.08$; $p = 0.003$). Finally, the single most significant predictor of RT in the GLM was R-trace ($t_{64} = 8.07$; $p < 0.001$). Therefore, quicker responding was not a function of specific choice-reward associations presented in a trial but of the GRS.

In sum, R-trace in our model encapsulates the persistence of previous unrelated rewards in memory, implements asymmetric and dynamic learning and affects reaction times. The dynamics of PEs emerging from this are consistent with the behaviorally observed pattern of stay/switch choices (Fig. 1) and create patterns of learning akin to spread of reward[11–13].

**Global reward state representation in Ia and DRN**. One of the main predictions from our winning RL model (RL + cl + cs + rt) is that the brain should hold a representation of the GRS. In a first GLM (GLM1) we indeed found such signals coding the R-trace variable from our winning model. We found the strongest evidence of encoding of R-trace in bilateral anterior insular cortex at the time of choice feedback (cluster-corrected at $Z > |2.6|$, $p = 0.05$), just posterior to OFC, which we refer to as agranular insula (Ia; Fig. 6a; Supplementary Table 1)[31]. It is notable that precisely this region of the macaque brain has recently been implicated in discrimination reversal learning[32] and may be important for balancing the influence of reward context versus specific reward outcomes. Consistent with this idea, we not only observed an effect of R-trace on bilateral Ia activity but also large signals, overlapping but slightly more posterior, reflecting specific outcome events (reward versus non-reward) (Fig. 6b). Both R-trace and outcome signals were positively signed which mirrored the positive effects that both R-trace and outcome had on PEs and therefore value updates in our RL model.

Another key feature of the model is that R-trace has a positive effect on choice value updates during both rewarded and unrewarded trials. This main effect makes positive PEs more positive and negative PEs less negative as a function of the GRS (Fig. 3c, d). For this reason, we examined, whether, similarly, Ia carries a positive representation of R-trace both during rewarded and unrewarded trials. We used a new GLM (GLM2) and applied a leave-one-out procedure to the previously identified peak coordinates. We extracted the effect sizes (COPE images) and performed a square-root transformation to de-weight outlying data points. We found that, strikingly, Ia represented R-trace during both rewarded and non-rewarded trials in the left (rewarded: $t_{24} = 2.07$; $p = 0.049$; unrewarded: $t_{24} = 3.68$; $p =$ 0.01) as well as in the right hemisphere (rewarded: $t_{24} = 3.44$; $p =$ 0.002; unrewarded: $t_{24} = 3.64$; $p = 0.001$; Fig. 6c, d). There was no significant difference between conditions in left ($t_{24} = 1.23$; $p =$ 0.23) or right Ia ($t_{24} = 0.73$; $p = 0.47$), nor did the effects differ between monkeys in the left (one-way ANOVA on averaged effect sizes: $F(3,21) = 1.206$; $p = 0.332$) or right Ia ($F(3,21) =$ 2.195; $p = 0.119$).

Ia, particularly left Ia (because of the extensive region in which strong R-trace and outcome effects overlapped), seemed to be the most likely brain region to integrate past and current rewards in a

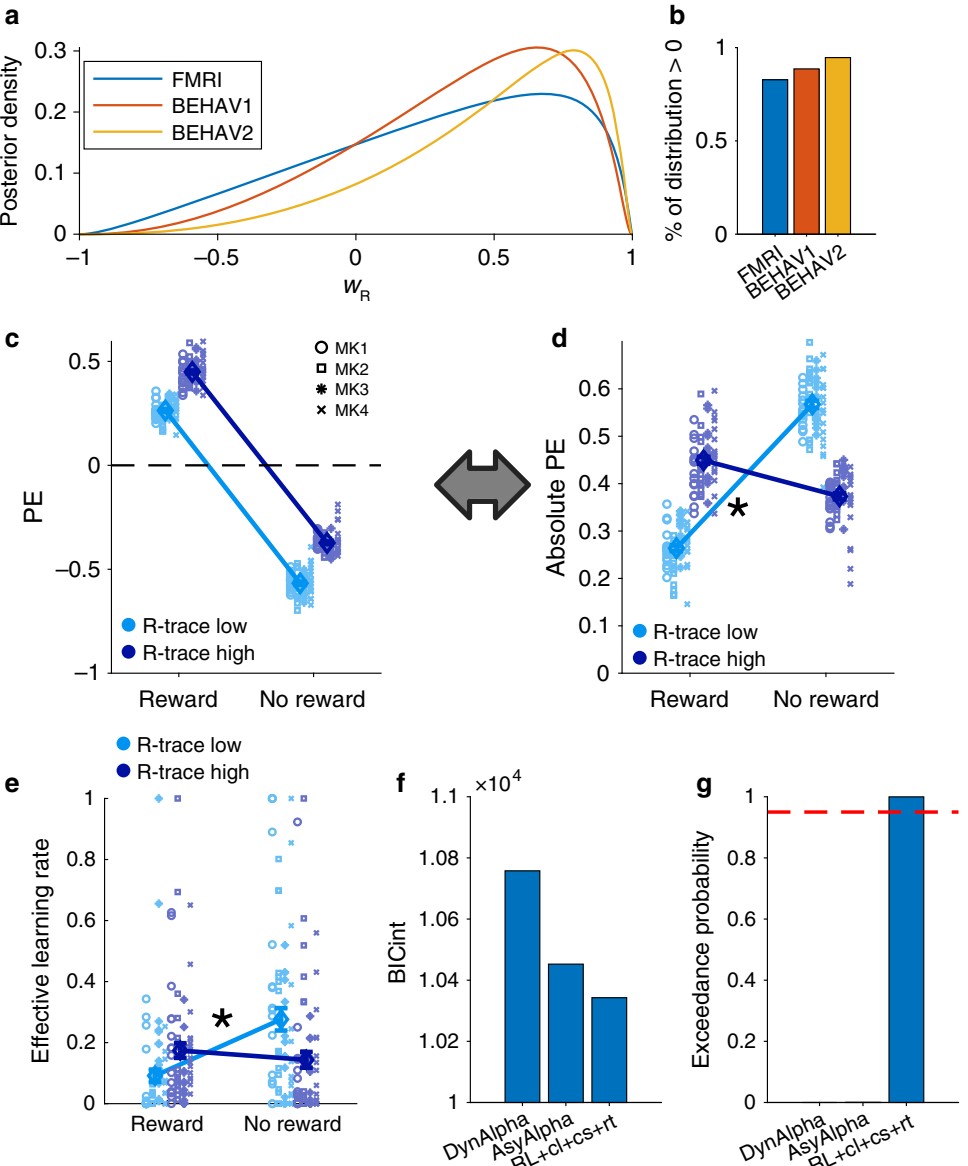

**Fig. 3 GRS asymmetrically and dynamically modulates the speed of learning. a** Posterior density over $w_R$ for the FMRI experiment as well as the two behavioral experiments. **b** Percentage of posterior distribution shown in panel **a** that is bigger than zero (83% for the fMRI experiment, and 87% and 95% for behavioral experiments 1 and 2, respectively). **c** Average PEs binned by current outcome (Reward/No reward) and R-trace (low/high; median split) show the offset of PE coding generated by positive $w_R$. **d** Absolute PEs plotted as in panel **c** illustrate stronger positive value updates when R-trace is high and stronger negative value updates when R-trace is low (ANOVA interaction effect; asterisk indicates $p < 0.001$). **e** Estimated effective learning rates of the model demonstrate that the speed of learning differs based on both the type of current outcome and the GRS (ANOVA interaction effect; asterisk indicates $p < 0.001$). Note the similarity to panel **d**. **f**, **g** BICint and exceedance probability indicate that the full model fits better than one using either asymmetric (AsyAlpha) or dynamic learning rates (DynAlpha). The dashed red line indicates an exceedance probability of 0.95. (in all panels except, error bars indicate ±SEM around the mean across all experimental sessions with $n = 65$). Source data are provided as Source Data file. Symbols in panel **c** indicate monkey identity in panels **c–e**; MK abbreviates monkey.

manner predicted by the computational model. In order to best understand neural activity in relation to behavioral and RL analyses, we regressed all effects of no interest out of the BOLD signal (i.e. all except R-trace and outcome effects) and then examined the residual BOLD signal (i.e. the variation in BOLD not explained by these regressors which we extracted from Matlab's stats.resid object). We found a striking temporal pattern of activity in left Ia when examining the residual BOLD time course in Ia as a function of both R-trace and outcome time-locked to the feedback onset (Fig. 7a–e). To this end, we binned it by R-trace (median split; low/high) and outcome (rewarded and unrewarded; R and no R) (compare with Fig. 3c). Initially,

residual BOLD activity clustered positively and negatively as a function of R-trace. However, 4–7 s after feedback onset (and thus at quite a late timepoint given that the monkey hemodynamic response function typically peaks after 3–4 s) Ia activity reflect the current outcome (reward or no reward; Fig. 7d). This became very clear by entering residual BOLD signals at every time point in a $2 \times 2$ ANOVA with the factors R-trace and outcome and examining the time course as a function of main effects and interaction. Highly consistent with our model predictions, we observed no interaction of R-trace and outcome at any point in the time course (Fig. 7e). We repeated the analysis for the right Ia and observed the same temporal pattern (Fig. 7f).

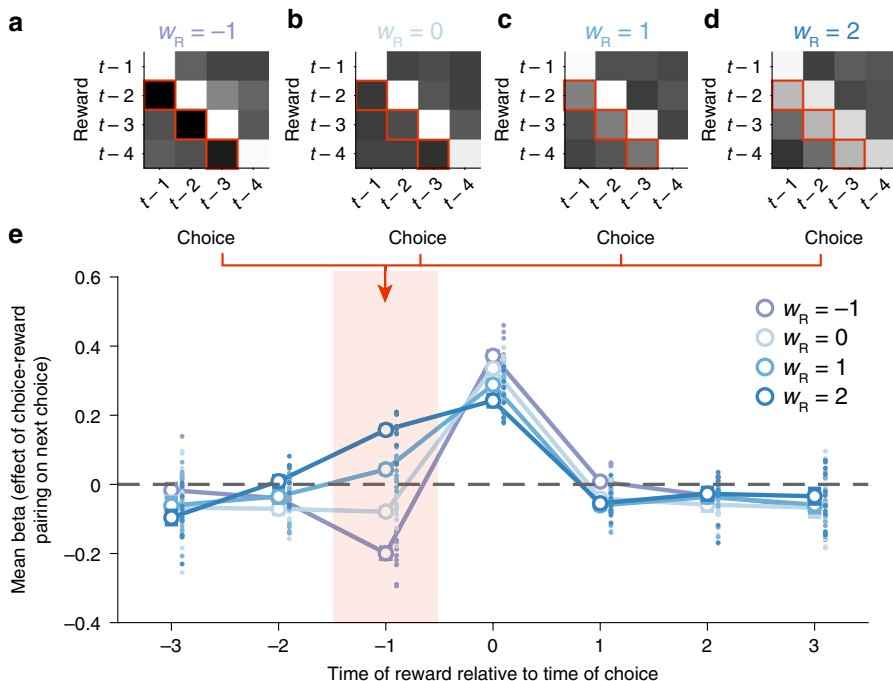

**Fig. 4 Model simulations show that forward spread of effect can emerge from the GRS. a–d** Choice simulations using different levels of $w_R$ (−1, 0, 1, 2, respectively) are analyzed with a GLM that estimates the effect of each possible choice-reward-pair going four trials back on subsequent choices. Lighter colors indicate more positive beta weights, i.e. subsequent choice probability is increased by reward delivered for the respective conjunction. For example, when $w_R$ is set to zero in panel **b**, the bright white diagonal indicates that the next choice the agent takes is influenced by the conjunction of choice and reward on the previous trial ($t$-1), the trial before that ($t$-2) and so on. However, as $w_R$ increases (**c**, **d**), the precise history of choice-reward conjunction has less effect and conjunctions between choices made on one trial and reward occurrences on other trials become more likely to influence the next choice that is taken (lighter colors at lower left in **c** and **d**). **e** Effects of rewards delivered at various time points relative to time point 0 on the probability of repeating the choice made at time point zero. Effects are derived from averaging beta weights shown in panels **a–d** along the diagonal with different offsets. The area shaded in red corresponds to the mean effects along the red squares in panels **a–d** and indicates that rewards have contrasting ($w_R < 0$) or facilitating effects on repeating subsequent choices ($w_R > 0$) depending on $w_R$. (error bars are ±SEM around the mean across simulated monkey data sets, $n = 12$).

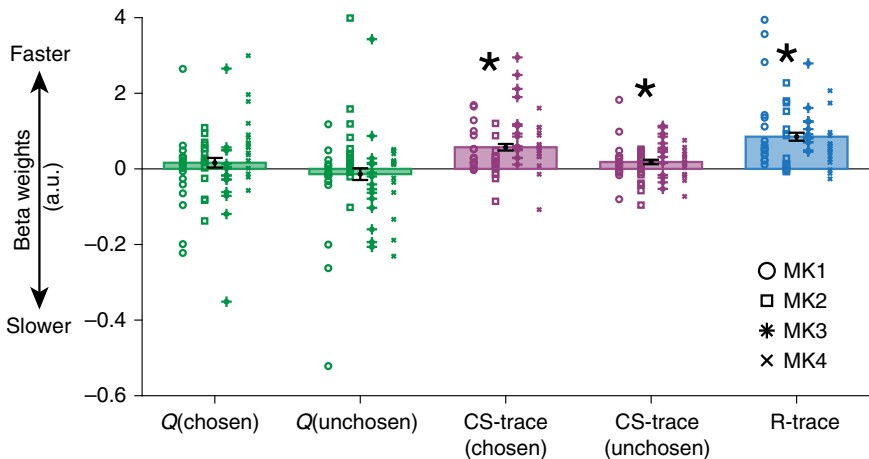

**Fig. 5 GRS impacts task engagement as measured by reaction times.** Beta weights of linear regression with the negative log of reaction times suggest that animals respond quicker when the choices that are offered on the current trial were recently pursued and also when R-trace is high (asterisks indicate $p < 0.001$, except for CS-trace (unchosen) where $p = 0.003$; we used two-sided one-sample $t$-tests against zero that remain significant also after Bonferroni correction for five comparisons). (error bars indicate ±SEM around the mean across sessions with $n = 65$). Source data are provided as Source Data file. Symbols indicate monkey identity; MK abbreviates monkey.

Note that we also found a significant effect of outcome when examining the right Ia time course at the time of the contralateral peak of this effect (time = 6.15 seconds; $F(1,24) = 7.2$; $p = 0.013$).

Complementary to our initial analysis, we repeated our GLM on the whole-brain level so that it was possible to identify areas that only represented R-trace when a trial was rewarded or when it was unrewarded. In the latter case we found a single cluster of activity outside Ia. The cluster was located in the brainstem with a clear peak in a location consistent with the DRN (Fig. 8a). Just as in Ia, overlapping within the same DRN region we also found

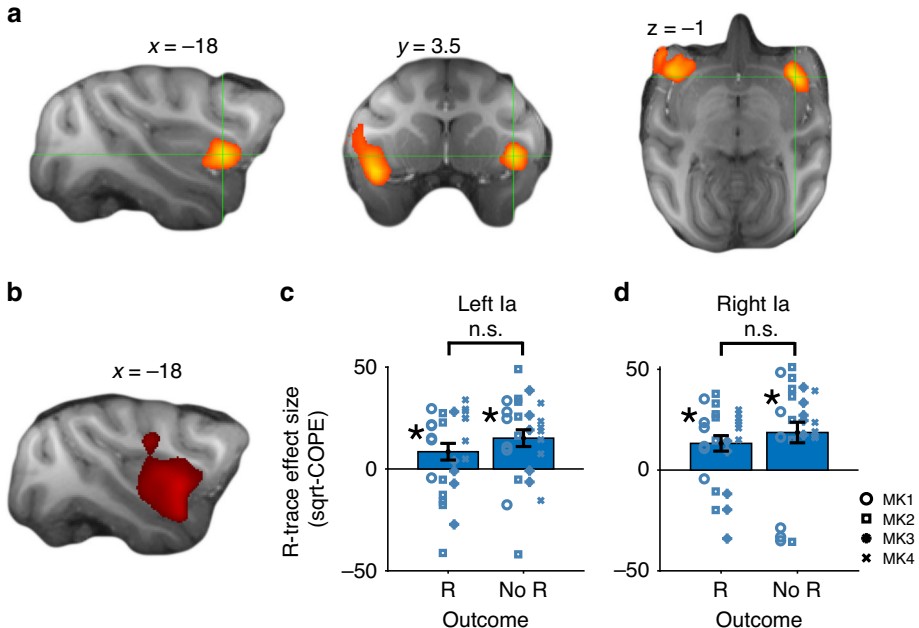

**Fig. 6 Bilateral Ia represents global reward state and current reward. a** Positive effects of R-trace bilaterally in Ia. **b** Both regions also showed concurrent overlapping or closely adjacent positive activation related to the current outcome (reward – no reward). **c, d** In accordance with our RL model, in both regions, R-trace exerted a positive effect during rewarded as well as unrewarded trials. These effects were revealed by two-sided one-sample $t$-tests against zero ($p$-values for left Ia are: $p(R) = 0.049$, $p(no\ R) = 0.01$; $p$-values for right Ia are: $p(R) = 0.002$, $p(no\ R) = 0.001$) and two-sided paired $t$-tests (difference in left Ia: $p = 0.23$ and right Ia: $p = 0.47$). ROIs were selected using a leave-one-out procedure. (Crosshairs highlight peak coordinates for subsequent ROI analysis; data are presented as mean values ±SEM across sessions; $n = 25$; dot color in **c, d** indicate monkey identity; all MRI results cluster-corrected at $Z > 2.6$, $p = 0.05$; asterisks indicate $p < 0.05$; n.s. indicates not significant). Source data are provided as Source Data file. Symbols indicate monkey identity in panels **c, d**; MK abbreviates monkey.

strong positive coding of current outcomes (region of interest (ROI) analysis: $t_{24} = 3.49$; $p = 0.002$). However, unlike Ia, DRN activity only reflected R-trace on non-reward outcome trials ($t_{24} = -3.028$; $p = 0.006$), but not on rewarded trials ($t_{24} = 0.51$; $p = 0.62$). There was a significant difference between conditions ($t_{24} = 2.96$; $p = 0.007$; Fig. 8b; using a leave-one-out DRN ROI to avoid bias). These effects of R-trace during unrewarded trials did not differ across monkeys: $F(3,21) = 2.403$; $p = 0.096$). We found some evidence for the opposite pattern as well as coding of current outcomes in dorsal anterior cingulate cortex (dACC; Supplementary Fig. 5). This is consistent with neurophysiological recordings demonstrating reward coding with multiple time scales in anterior cingulate cortex[33–36].

As expected from the whole-brain analyses, the time courses of effects in DRN and dACC looked strikingly different to Ia and there were outcome x R-trace interaction effects at some points (Fig. 8c, d; Supplementary Fig. 5a–e). Both DRN and dACC showed significantly earlier outcome encoding compared to left and right Ia (Supplementary Fig. 5f). Therefore, while DRN and dACC represent outcomes at the time of feedback, Ia's neural activity is better described as reflecting ongoing integration of new rewards into a longer-term representation of the choice-unlinked value of each animal's global state.

**Coding of value and choice history in prefrontal cortex.** In the above sections we reported activity reflecting GRS and outcome signals during learning. In this final section, we examine activity reflecting the decision variable that resulted from this learning process and that combines choice values (informed by both the GRS and choice contingent reward learning) and reward-unlinked history of past choices. We tested whether any brain region integrated choice evidence in the same comprehensive manner as was apparent in behavior. We regressed the BOLD

signal against the DV (coded as chosen – unchosen; $DV_{total}$ of the full model RL + cl + cs + rt) at the time of decision (GLM3). We found a large cluster of activity focused in vmPFC/mOFC but spreading into adjacent parts of prefrontal cortex (Fig. 9a). As in previous reports, these effects were negatively signed (i.e. negative relationship between higher relative choice evidence and BOLD signal)[20,37].

One possibility is that this activity is solely driven by the relative reward expectation associated with the chosen option because this is part of $DV_{total}$. Alternatively, prefrontal activity might reflect this but in addition integrate the weight that the previous history of choices has on the subsequent decision. To address this question, we set up a new GLM (GLM4) that broke $DV_{total}$ apart into its two component parts, $DV_{value}$ and $DV_{choice}$. $DV_{value}$ is the difference in Q-value between chosen and unchosen options and $DV_{choice}$ is the remaining part of $DV_{total}$ and comprises the weighted choice-location and choice stimulus traces (CL-trace and CS-trace, Fig. 2d). Note that $DV_{choice}$ is a measure of choice persistence; positive values of this variable indicate that the chosen option was recently more often picked than the unchosen one, and negative values indicate the opposite. For $DV_{value}$, we again found a cluster of activity in vmPFC/mOFC that overlapped with $DV_{total}$ (Fig. 9b). However, the effects were considerably smaller than for the full decision variable. To test this difference formally, we performed a neural model comparison in vmPFC[20] using a leave-one-out procedure to avoid bias (see Methods section). We regressed vmPFC BOLD against two identical GLMs with the only difference that one of them included $DV_{total}$ and the other one $DV_{value}$ instead. We found that that the exceedance probability favored the $DV_{total}$ model by an extensive margin (exceedance probability = 0.999; Fig. 9c). This suggests that vmPFC/mOFC integrates different types of evidence such as reward expectation and choice history into a

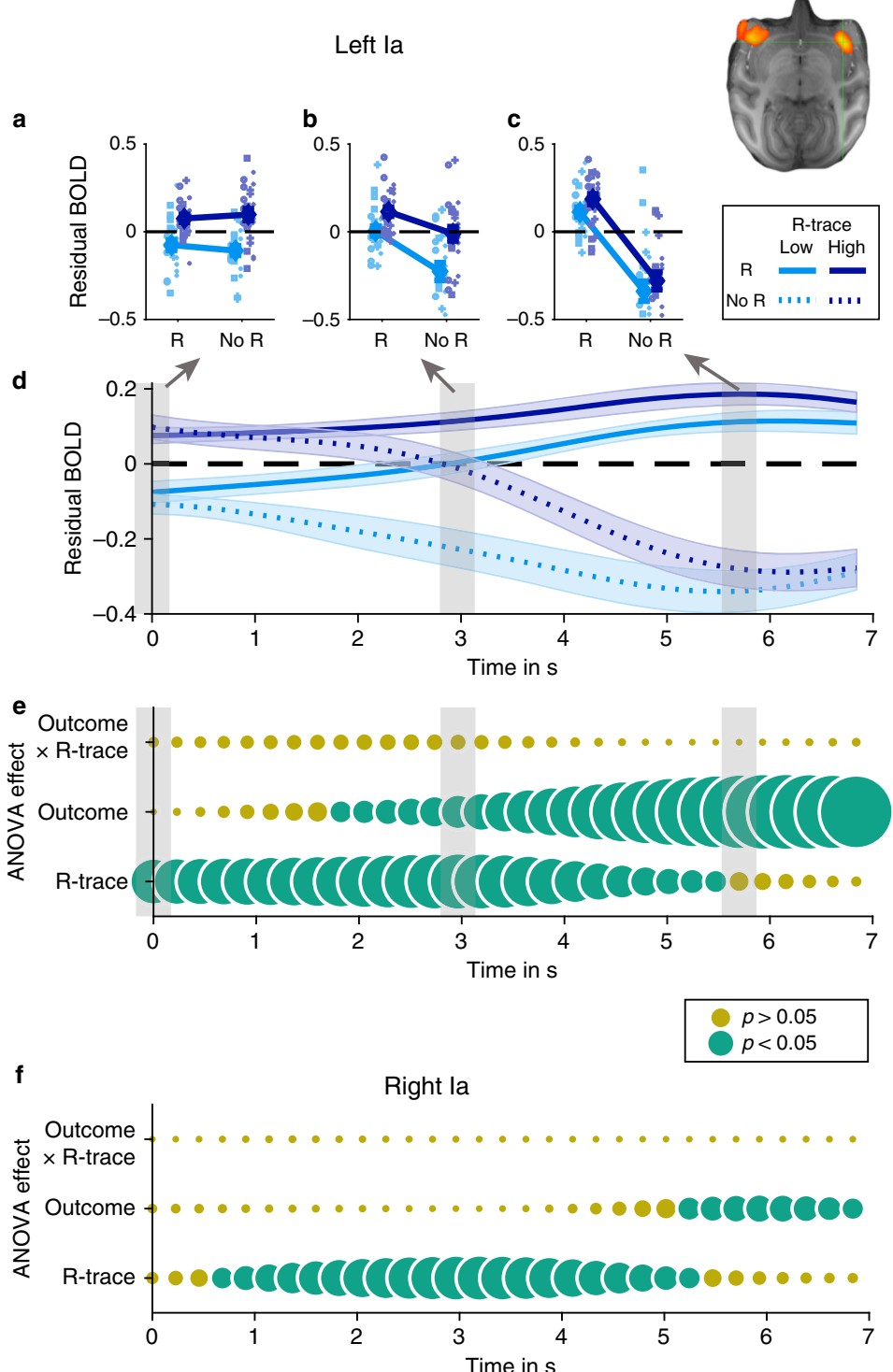

**Fig. 7 Timecourse of bilateral Ia activity. a–c** Residual BOLD effects in left Ia binned by R-trace (low/high; median split) and outcome (reward/no reward: R and no R) at three time points after feedback delivery (indicated as gray bars in panel **d**; feedback occurs at time 0 s and the approximate peak of the macaque hemodynamic response function is 3–4 s later). **d** Time course of the four effects shown in panels **a–c**. Note that the BOLD signal clusters initially as a function of R-trace (low/high) and then regroups as a function of outcome (no reward/reward). Zero is time of feedback onset. **e** F-statistic for 2 × 2 (R-trace x outcome) ANOVA applied at each time step to the data shown in panel **d**. Rows indicate main effects and interaction effect. Circles are scaled F-values. Larger circles indicate higher F-values. Yellow indicate p-values of effects are bigger than 0.05. Green circles indicate effects for a given row and time point where $p < 0.05$. **f** Same analysis as in panel **e** applied to the right Ia replicate pattern of effects found in left Ia. Note that panels **e, f** (and subsequent analogous panels) are for illustration as this analysis does not control for autocorrelations in the BOLD signals and multiple comparisons. (Data are presented as mean values ±SEM across sessions; $n = 25$). Symbols indicate monkey identity in panels **a–c**.

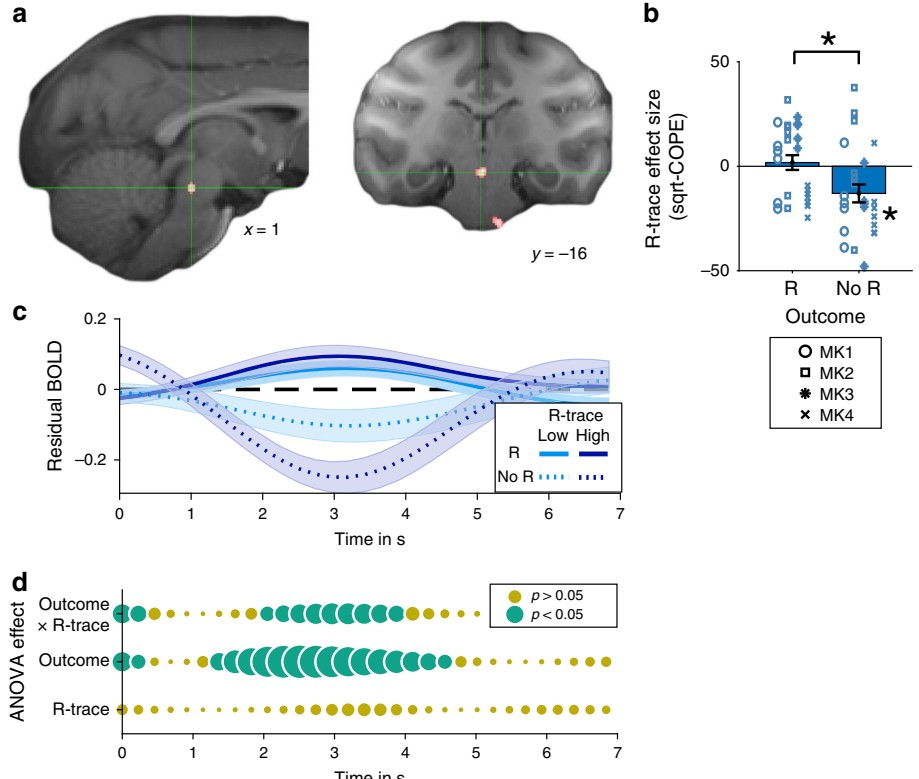

**Fig. 8 R-trace and outcome signals in DRN. a** During unrewarded trials, DRN encoded R-trace negatively. This effect is shown at $z > 3.3$ for spatial specificity. Reward feedback occurs at time 0 s and the approximate peak of the macaque hemodynamic response function is 3–4 s later. **b** This was not the case during rewarded trials (reward/no reward: R and no R). We used two-sided paired t-tests to assess the difference in activation ($p = 0.007$). The asterisk for no R refers to the whole-brain analysis performed in panel **a**. ROIs were selected using a leave-one-out procedure. **c, d** Time course analyses of DRN activity related to outcome and R-trace coding. Same conventions as in Fig. 7. (Crosshairs highlight peak coordinates for subsequent ROI analysis; data are presented as mean values ±SEM across sessions; $n = 25$; dot color in **b** indicates monkey identity; all MRI results cluster-corrected at $Z > 2.6$, $p = 0.05$; asterisks indicate $p < 0.05$). Source data are provided as Source Data file. Symbols indicate monkey identity in panel **b**; MK abbreviates monkey.

compound decision variable rather than comparing reward associations alone (Supplementary Fig. 6).

Lastly, we compared the signatures of reward processing across our three ROIs: Ia (bilaterally), DRN and vmPFC. While non-contingent GRS signals were confined to Ia and DRN, value comparison signals were vice versa specific to vmPFC (Fig. 10, Supplementary Fig. 7).

## Discussion

Animals' choices are driven by past choice-reward conjunctions. Here, we find that they are also heavily influenced by the global reward state (GRS) as well as by the history of past choices per se. Our analysis of the macaque BOLD suggests critical roles for Ia, DRN, and vmPFC in this process.

Our behavioral analyses conceptualize the macaque's choices as a stay/leave-type of decision[21–23] and show that animals have a strong tendency to repeat rewarded choices, but also, more generally, simply to repeat previous choices regardless of reward. In addition, the GRS had a very specific effect on behavior: regardless of a choice's specific history of reward association, animals tended to stay with choices made when GRS was high but increasingly explored alternative options when GRS was low (Fig. 1).

Our winning RL model formalized learning mechanisms underlying these effects and thereby supported and extended our behavioral findings (Figs. 2–4). The model is inspired by previous models considering the average reward rate[15,17,18,28]. It uses an estimate of the GRS (i.e. R-trace) to bias PE calculation and

facilitate positive value updates in high GRS but relatively depress them in low GRS. A key feature of the model is that the direction of influence of the GRS on learning is determined empirically by fitting $w_R$; it can be positive or negative (or absent). The positive effects of GRS observed in our study are in line with previous reports of spread of effect[8,11–13] and therefore suggest a mechanism by which the (mis-) assignment of previous rewards to subsequent choices can emerge within an RL framework. The GRS may act a as a proxy for how good the current and expected states of the animal are and this might bias the evaluation of newly encountered options. Consistent with this interpretation, GRS not only influenced learning but was also tightly linked to the overall readiness of the animal to engage in the task as measured by the response times (Fig. 5). In this view GRS is reminiscent of future reward expectations used in multi-state applications of temporal difference algorithms[4,38]. While such learning mechanisms may be ineffective in laboratory bandit-tasks, they may well be adaptive in the natural environments in which animals and humans have evolved, where choices often have serial dependencies[18,21,39]. Negative $w_R$, on the other hand, implements a temporal contrast effect whereby the value of new outcomes is de-weighted in a high GRS, because they are referenced to the already high value of the environment. Such contrast effects have been reported in other contexts[17,18,28,40]. Therefore, our model suggests a common algorithmic implementation of a variety of GRS related effects. The model, moreover, captures modulations of the speed of learning by the GRS. While the GRS influences learning via a modulation of learning rates in other

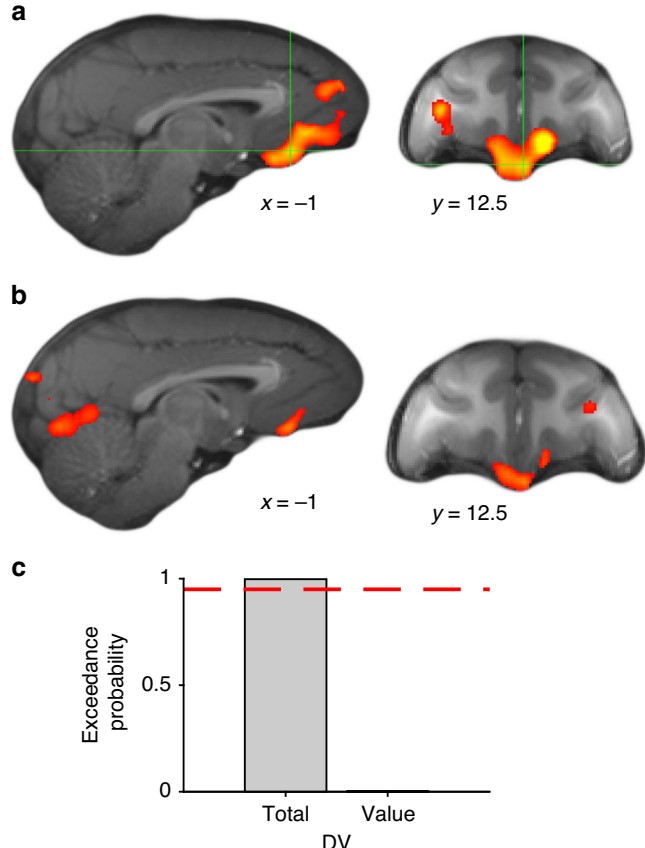

**Fig. 9 Decomposition of vmPFC and DRN activity in value and choice.**
**a** Effects of DV$_{total}$ in vmPFC/mOFC. **b** DV$_{value}$, the part of DV$_{total}$ reflecting the value difference between chosen and unchosen options, correlated with a similar vmPFC area, but signal sizes were weaker. **c** Bayesian model comparison suggests vmPFC activity is better described by DV$_{total}$ than by DV$_{value}$. ROIs were determined using a leave-one-out procedure. The dashed red line indicates an exceedance probability of 0.95 ($n =$ 25 sessions; all MRI results cluster-corrected at $Z > 2.6$, $p = 0.05$). Source data are provided as Source Data file.

models[41], it affects value updates directly in our model. This means that it can reproduce dynamic changes in value update rates and asymmetries in value updates without explicitly adjusting learning rates. Our model also accords well with optimism biases observed in human learning[42,43]. Assuming optimism is reflected in the belief of a general high GRS, our model predicts that optimists weight positive outcomes more strongly than negative ones and can even ignore negative feedback.

Our neural results suggest a critical role for Ia[31] in reward learning (Figs. 6 and 7). Unlike for example dACC[34–36,44], Ia is rarely targeted in neurophysiological studies of reward-guided learning or decision making. However, reward-related responses have been observed in macaque Ia using fMRI[45] and human fMRI studies provide evidence of rewards being represented with multiple time constants in Ia[18,46]. Extending these findings, we found macaque Ia BOLD signals to carry information about both current and past outcomes (i.e. the GRS) simultaneously and these signals closely mimicked the way by which our model combined the influence of reward context and specific reward outcomes to guide value updates. Thes results suggest a way to begin reconciling longstanding debates about the role of adjacent OFC in mediating behavioral flexibility particularly during reversal learning tasks. Choice-reward discrimination reversal learning is impaired after excitotoxic lesions of OFC in rodents and marmosets[47–50] - but not in macaques[51]. One possibility that might resolve this species discrepancy is that a critical region for mediating behavioral flexibility is, or is homologous to, macaque Ia. The idea that the GRS and outcome information carried by macaque Ia might be critical for reversal learning is further supported by the three observations. First, recently, links between macaque Ia and rodent OFC[52] and possibly marmoset OFC[53] have been highlighted. Second, macaque Ia - and not adjacent OFC - exhibits most extensive gray matter change during choice-reward discrimination reversal learning[32]. Finally, aspiration lesions of macaque OFC disrupt choice-reward discrimination reversal learning[51] and have a profound impact on Ia. Future neurophysiological and lesion studies targeting Ia could help clarify the contribution of reward computations in Ia to behavioral flexibility.

The fact that serotonin depletion in marmoset orbitofrontal cortex[54,55] also disrupts reversal learning is also consistent with

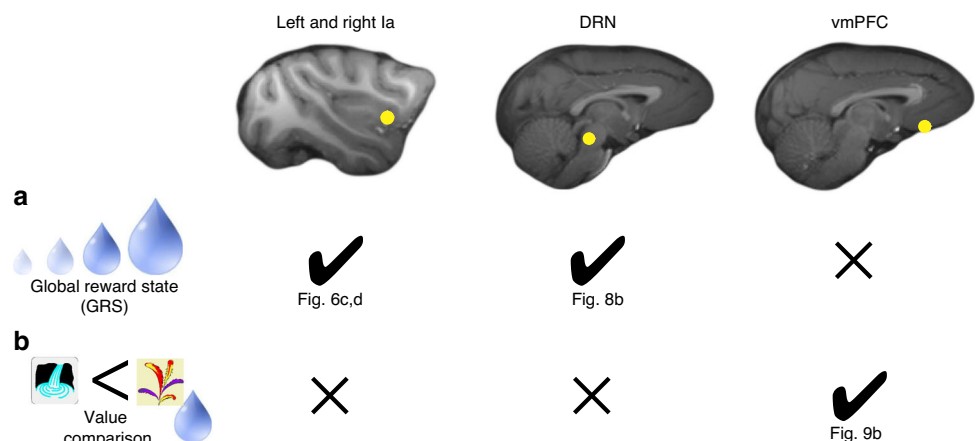

**Fig. 10 Computations of learning and choice in Ia, DRN, and vmPFC.** Check marks indicate observed effects with references to the respective figure panel. X indicates absent effects as detailed in Supplementary Fig. 7. **a** Bilaterally in Ia as well as in DRN, we found a representation of the GRS, i.e. a choice-unlinked general representation of recent reward. We did not observe such a non-contingent general reward signal in vmPFC. **b** In vmPFC, however, we found a value difference signal capturing the difference in contingent reward expectation of the chosen compared to the unchosen option (DV$_{value}$). These signals were clearly absent in Ia and DRN. The activity patterns in Ia and DRN suggest they are more related to learning than to decision making. (leftmost symbols used in the figure are adapted from Chau et al.[19].

the present finding that both Ia and DRN – an origin of serotoninergic innervation of cortex – possess complementary GRS signals. DRN neurons in rodents encode positive and negative PEs[56] and manipulation of DRN activity can affect the learning rate by which observed outcomes impact value updates[57]. Such up- and down-regulation of the impact of newly observed outcomes is apparent in the BOLD signal we observe during negative outcomes in DRN. The negative response to reward omission appears to be even more negative in states of high GRS. This suggests new ways of thinking about the impact of serotonergic treatments in the clinic. Rather than simply reflecting the likelihood of behavioral inhibition per se, activity in DRN reflect how an adverse event will be interpreted in the light of the current reward context and it is this combination of factors – the recent outcome and the GRS – that determines whether change in behavior will occur.

Finally, vmPFC/mOFC integrated stimulus-reward associations as well as choice history into a compound DV. Behaviorally, the animals picked high-reward options but also repeated choices regardless of reward delivery. The neural signals in vmPFC closely mirrored this choice pattern. BOLD signals were stronger when animals opted for high value choices and also when they opted for previously chosen choices regardless of reward. This meant that vmPFC signaled the overall readiness of the animal to engage in a particular choice, may this be caused by previous experiences of reward or because the option was in the focus of attention in the recent past[58,59].

## Methods

**Subjects**. Four male rhesus monkeys (Macaca mulatta) were involved in the experiment. They weighed 10.4–11.9 kg and were 7 years of age. They were group housed and kept on a 12-h light dark cycle, with access to water 12–16 h on testing days and with free water access on non-testing days. All procedures were conducted under licenses from the United Kingdom (UK) Home Office in accordance with the UK The Animals (Scientific Procedures) Act 1986 and with the European Union guidelines (EU Directive 2010/63/EU).

**Behavioral training**. Prior to the data acquisition, all animals were trained to work in an MRI compatible chair in a sphinx position that was placed inside a custom mock scanner simulating the MRI scanning environment. They were trained to use custom-made infrared touch sensors to respond to abstract symbols presented on a screen and learned the probabilistic nature of the task until reaching a learning criterion. The animals underwent aseptic surgery to implant an MRI compatible head post (Rogue Research, Mtl, CA). After a recovery period of at least 4 weeks, the animals were trained to perform the task inside the actual MRI scanner under head fixation. The imaging data acquisition started once they performed at >70% accuracy (choosing the option with the highest expected value) for at least another three consecutive sessions in the scanner.

**Experimental task**. Animals had to choose repeatedly between different stimuli that were novel in each testing session (Fig. 1). Each session comprised 200 trials. We used a probabilistic reward-based learning task. The task consisted of a series of choices, on each trial, between two stimuli drawn out of a larger pool of three. The position of the two available options on the left and right side of the screen were pseudorandomized. Animals had to choose any symbols by touching one of two infrared sensors placed in front of their two hands corresponding to the stimuli on the screen. After making their decision, if the correct option was selected, the unselected option disappeared, and the chosen option remained on the screen and a juice reward was delivered. If an incorrect choice was made, no juice was delivered. The outcome phase lasted 1.5 s. Each reward was composed of two 0.6 ml drops of blackcurrant juice delivered by a spout placed near the animal's mouth during scanning. The experiment was controlled by Presentation software (Neurobehavioral Systems Inc., Albany, CA). We used an intertrial-interval of 5–7 s for the MRI experiment, but there was no temporal jitter between decision and outcome phase (except reaction time) to keep the animals engaged in the task.

The experimental task was the same in all three experiments as were the four monkeys, but data collection for the experiments was separated by several months and acquired over the course of a 2-year period. For the MRI experiment, each animal performed five to seven sessions (25 sessions overall). For the two behavioral experiments, each animal performed five sessions (20 sessions per behavioral experiment).

**Behavioral data analysis**. We ran logistic general linear model (GLM) analyses, implemented in Matlab 2018a version 9.4.0, on the behavioral choice data. To be able to estimate complex GLMs with a large regressor set, we concatenated experimental sessions per animal per experiment and applied the GLM to the concatenated sessions (Fig. 1f, g; Supplementary Fig. 1b, c; Fig. 4 uses the same procedure with simulated choice data). Note that Supplementary Fig. 1 describes all 22 regressors used in our main behavioral GLM. As all four animals participated in all three experiments, this procedure resulted in 12 beta weights (per regressor) overall, three per animal. We tested these resulting beta weights for significance using analyses of variance (ANOVA) and one-sample $t$-tests against zero, implemented in Matlab and Jasp version 0.9.0.1. For GLMs with smaller regressor sets effect sizes were estimable on the basis of single sessions. In such a case we applied the GLMs separately for each of the 65 sessions collapsed over experiments, for instance for the reaction time GLMs. Reaction time analyses used a linear link function. Beta weights were tested for significance using and one-sample $t$-tests against zero.

**Basic RL architecture**. All reinforcement learning (RL) models shared the same basic architecture. Three value estimates, $Q(A)$, $Q(B)$, and $Q(C)$ tracked the rewards received for choosing each of the three stimuli that were presented in each session. Note that each session used new stimuli to avoid carry over effects. All $Q$-values were initialized at 0.5 at the start of each session and the chosen stimulus was updated based on the discrepancy between outcome (0 or 1 for reward and no reward, respectively) and $Q$-value, scaled by a learning rate $\alpha$. Unchosen $Q$-value estimates remained unchanged. For example, if option $A$ was chosen on trial $t$, then its prediction error (PE) and value update would be calculated based on the reward $r$ as follows:

$$\text{PE}_t(A) = r_t - Q_t(A) \tag{1}$$

$$Q_{t+1}(A) = Q_t(A) + \alpha \text{PE}_t(A) \tag{2}$$

For all models, the decision variable (DV) reflected the evidence for making a rightward choice. Note that the identity of the left and right choice (whether they were option $A$, $B$ or $C$) was pseudorandom. For the simplest model, RLsimple, which did not include any choice or reward memory traces, the DV for each trial was simply calculated as the value of the stimulus presented on the right side minus the value of the stimulus presented on the left side plus an additional additive constant reflecting a side bias (SB).

$$\text{DV}_{\text{RLsimple},t} = Q_t(\text{right}) - Q_t(\text{left}) - \text{SB} \tag{3}$$

Then, for any model $m$ (here $m = \text{RLsimple}$), the decision variable was filtered through a standard softmax function to calculate the probability of a rightward choice.

$$p_t(\text{right}) = \frac{1}{1 + e^{-\beta \times \text{DV}_{m,t}}} \tag{4}$$

The probability of the observed choice on each trial was calculated as:

$$p_t(\text{choice}) = \begin{cases} p_t(\text{right}), & \text{if right option is chosen} \\ 1 - p_t(\text{right}), & \text{otherwise} \end{cases} \tag{5}$$

The simplest model, RLsimple, comprised three free parameters: the learning rate $\alpha$ (bound between 0 and 1), the inverse temperature parameter $\beta$ (bound between 0 and positive infinity) and the side bias SB (bound between −1 and 1). Subsequent models shared the basic architecture of model RLsimple and comprised the same free parameters with the same bounds plus additional free parameters. Models including choice traces modified the basic DV by adding terms related to the history of past choice stimuli and choice locations. By contrast, models including a reward trace modified the calculation of prediction errors.

**Modeling of memory traces**. All memory traces were initialized at zero at the start of a session and updated in the following ways. The choice-location trace (CL-trace) was updated on every trial based on the discrepancy between the actual choice-location ($L$), coded as −1 or 1 (for right and left side, respectively), and the CL-trace, scaled by a learning rate $\alpha_{\text{CL}}$.

$$\text{CL-trace}_{t+1} = \text{CL-trace}_t + \alpha_{\text{CL}}(L_t - \text{CL-trace}_t) \tag{6}$$

The choice stimulus trace (CS-trace) decayed exponentially from one trial to the next one with a given rate determined by a free parameter $\lambda_{\text{CS}}$[4,27]. However, the CS-trace for the chosen option was set to 1 at the end of the trial. For example, the decay of the CS-trace for a stimulus $A$ on trial $t$ was calculated as:

$$\text{CS-trace}_t(A) = \lambda_{\text{CS}}\text{CS-trace}_{t-1}(A) \tag{7}$$

The reward trace (R-trace) was updated on every trial based on the discrepancy between R-trace and the observed outcome, scaled by a learning rate $\alpha_{\text{R}}$.

$$\text{R-trace}_{t+1} = \text{R-trace}_t + \alpha_{\text{R}}(r_t - \text{R-trace}_t) \tag{8}$$

Note that the R-trace calculation was independent of the specific choices taken and hence only knowledge about the factual sequence of outcomes was required to calculate R-trace. Similarly, CL-trace and CS-trace required only knowledge of the choice location and choice stimulus, respectively, ignoring the sequence of outcomes experienced over the course of a session. $\lambda_{\text{CS}}$, $\alpha_{\text{CL}}$, and $\alpha_{\text{R}}$ were all bound between 0 and 1.

All three memory traces exerted their influence on learning and choice scaled by weight parameters. In all cases, the weight parameters could be positive, zero, or negative, meaning that the magnitude and direction of influence of the memory traces were determined empirically during model fitting. CL-trace and CS-trace were added to the DV. The CL-trace could be added directly, since it was already coded in terms of spatial location similarly to the DV itself (although inverted). For the CS-trace, the influence on choice was determined by the difference in CS-trace between right and left stimulus. Note that the CS-trace difference was added to the DV after the trial-wise decay but before the update of the chosen stimulus. Below the DVs for models that contained only one of the two or both choice traces in the DV in addition to the components from RLsimple.

$$\text{CL-bonus}_t = -\text{CL-trace}_t \tag{9}$$

$$\text{CS-bonus}_t = \text{CS-trace}_t(\text{right}) - \text{CS-trace}_t(\text{left}) \tag{10}$$

Then for Models CL, CS, and CL+CS, we have respectively:

$$\text{DV}_{\text{CL},t} = \text{DV}_{\text{RLsimple},t} + w_{\text{CL}}\text{CL-bonus}_t \tag{11}$$

$$\text{DV}_{\text{CS},t} = \text{DV}_{\text{RLsimple},t} + w_{\text{CS}}\text{CS-bonus}_t \tag{12}$$

$$\text{DV}_{\text{CL+CS},t} = \text{DV}_{\text{RLsimple},t} + w_{\text{CL}}\text{CL-bonus}_t + w_{\text{CS}}\text{CS-bonus}_t \tag{13}$$

Note that positive values of $w_{\text{CL}}$ indicate that the DV of the current trial would be biased towards the same location as the direction of previous choices, and positive values of $w_{\text{CS}}$ indicate that the DV would be swayed towards the stimulus with the highest CS-trace. In other words, positive values of $w_{\text{CL}}$ and $w_{\text{CS}}$ reflect a tendency to repeat predominant previous choice locations and predominant previous choice stimuli, respectively.

In any model $m$ that included R-trace, R-trace was inserted directly into the calculation of the prediction error scaled by a weight parameter $w_{\text{R}}$, which was allowed to range between $-1$ and 1. For example, if option $A$ was chosen on trial $t$, the corresponding PE would be calculated as:

$$\text{PE}_{m,t}(A) = r_t + w_{\text{R}}\text{R-trace}_t - Q_t(A) \tag{14}$$

**Model fitting**. All RL modeling was conducted in Matlab (version 2018a). We used an iterative expectation-maximization (EM) algorithm to fit the models[25]. We estimated models both over all experiments and for each experiment and monkey separately. This ensured that our results were valid over all data sets but also within each experiment and monkey. It also enabled us to use the experiment-appropriate parameter estimates for the analysis of the MRI data. In all cases the fitting comprised two levels: the lower level of the individual sessions and the higher-level reflecting either all of the sessions together, all sessions from the same experiment, or all sessions from the same monkey.

During the expectation step, we calculated the log-likelihood of the subject's series of choices given a model M and its parameter vector $\mathbf{h}_i$ of each session $i$ ($i \in \{1..N\}$). To do so, we summed the conditional probability of each trial's choice given the model's DV and parameters $\mathbf{h}_i$ (here abbreviated $\log(p(\text{choice}_t|\text{DV},\mathbf{h}_i))$) over all trials of a session. We then computed the maximum posterior probability (PP$_i$) estimate obtained with this parameter vector $\mathbf{h}_i$, given the observed choices and given the prior computed from group–level Gaussian distributions over the parameters with a mean vector $\boldsymbol{\mu}$ and standard deviation $\boldsymbol{\sigma}^2$.

$$\mathbf{PP}_i = \max_h\left[\sum_t(\log(p_t(\text{choice}_t|\text{DV},\mathbf{h}_i)) + \log(\text{normpdf}(\mathbf{h}_i|\boldsymbol{\mu},\boldsymbol{\sigma})))\right] \tag{15}$$

$$\mathbf{h}_i = \arg\max_h\left[\sum_t(\log(p_t(\text{choice}_t|\text{DV},\mathbf{h}_i)) + \log(\text{normpdf}(\mathbf{h}_i|\boldsymbol{\mu},\boldsymbol{\sigma})))\right] \tag{16}$$

We initialized the group–level Gaussians as uninformative priors with means of 0.1 (plus some added noise) and variance of 100. During the maximization step, we recomputed $\boldsymbol{\mu}$ and $\boldsymbol{\sigma}$ based on the estimated set of $\mathbf{h}_i$ and their Hessian matrix $\mathbf{H}_i$ (as calculated with Matlab's fminunc) over all $N$ sessions.

$$\boldsymbol{\mu} = \frac{1}{N}\sum_i \mathbf{h}_i \tag{17}$$

$$\boldsymbol{\sigma}^2 = \frac{1}{N}\sum_i[\mathbf{h}_i^2 + \text{diag}(\text{pinv}(\mathbf{H}_i))] - \boldsymbol{\mu}^2 \tag{18}$$

where the diagonal terms of the inverted Hessian matrix (computed in Matlab with diag(pinv($\mathbf{H}_i$))) give the second moment around $\mathbf{h}_i$, approximating the variance, and thus the inverse of the uncertainty with which the parameter can be estimated[25]. We repeated expectation and maximization steps iteratively until convergence of the posterior likelihood PP$_i$ summed over the group or a maximum of 800 steps. Convergence was defined as a change in PP$_i < 0.001$ from one iteration to the next. Note that bounded free parameters (for example the learning rates) were transformed from the Gaussian space into the native model space via appropriate link functions (e.g. a sigmoid function in the case of the learning rates) to ensure accurate parameter estimation near the bounds.

**Model comparison**. We compared fitted models by calculating their integrated BIC (BIC$_{\text{int}}$)[28]. For this, we drew $k = 2000$ samples of parameter vector $\mathbf{h}_i$ per session $i$ from the Gaussian population distributions using the final estimates of $\boldsymbol{\mu}$ and $\boldsymbol{\sigma}$, and computed the negative log likelihood (NLL$_{i,k}$) of each sample and session using the equation (corresponding to the first part in Eq. (15))

$$\text{NLL}_{i,k} = -\sum_t \log\left(p_t\left(\text{choice}_t|\mathbf{h}_{i,k}\right)\right) \tag{19}$$

Next, we integrated the NLL$_{i,k}$ over samples $k$ and sessions $i$ and calculated BIC$_{\text{int}}$ based on the integrated log-likelihood (iLog) in the following way:

$$\text{iLog} = \sum_i \log\left(\sum_{k=1}^{2000} e^{-\text{NLL}_{i,k}}/2000\right) \tag{20}$$

$$\text{BIC}_{\text{int}} = -2 \times \text{iLog} + \text{Np} \times \log\left(\sum \text{Nt}_i\right) \tag{21}$$

Np refers to the number of free parameters per model and Nt$_i$ refers to the number of trials per session $i$.

As a second index of model fit, we used the Laplace approximation to calculate the log model evidence (LME) per session $i$ based on the posterior probability PP$_i$ (see Eq. (15)):

$$\text{LME}_i = -\text{PP}_i - \frac{1}{2}\log(\det(\mathbf{H}_i)) + \frac{\text{Np}}{2}\log(2\pi) \tag{22}$$

Note that the hessian matrix was calculated based on the posterior estimates using the likelihood and the initial group–level prior estimates. We submitted the LME scores to spm_BMS[29] to compute the exceedance probability, the posterior probability that one model is the most likely model used by the population among a given set of models. In addition, we computed the session-wise difference in LME between two candidate models to approximate log Bayes factors, i.e. the ratio of posterior probability of the models given the data[60,61].

**Supplementary control models**. In our main model comparison, which follows on from our behavioral analyses, we formalize the effects of non-contingent choice and reward memories on decision making in an RL framework. We consider models with a choice-location trace (CL-trace), a choice stimulus trace (CS-trace), and a reward trace (R-trace) that modulates PEs. We consider all possible permutations of these mechanisms and find that a model including all three memory traces (RL + cl + cs + rt) fits the observed behavior best. We refer to this model also as the full model. However, in a supplementary model comparison we also consider two additional categories of control models.

The first category assumes, just as the full model, an effect of R-trace, but one that is conveyed not via a modulation of PEs but instead via affecting non-value-related mechanisms such as the CL-trace or CS-trace, or the $\beta$ parameter of the softmax function (Eq. (4)). Such a modulation of choice memories could in principle generate behavior similar to the one we had observed – that animals increasingly stay with rewarded choices in a high GRS, but preferably switch away from options in low GRS environments. Similar effects are possible if R-trace modulated the $\beta$ parameter because it might increase or decrease random exploration at times of low or high GRS. Such modes of action of R-trace are possible, albeit somewhat unlikely in relation to the choice traces. This is because, when reward trace exerts its impact via the PE calculation its effects remain within the realm of value in the full model. However, the alternative account would imply that non-contingent reward memories can only be pressed into action in interaction with non-contingent choice memories. While this is not impossible, it would require us to always first assume that decision making is influenced by non-contingent choice memories and that additional effects of non-contingent reward traces are always secondary to and mediated by these choice memories. Nevertheless, we tested a comprehensive set of alternative models that incorporated such R-trace mechanisms. These alternative models mimicked the full model but eliminated the effect of R-trace from the PE calculation (Eq. (14)) and thereby also removed the free parameter $w_{\text{R}}$ that determined the weight of R-trace on the PE. Instead, in the alternative models, R-trace modulated the weights of CL-trace, CS-trace or both. We denote the new models by the interaction type of R-trace with the respective choice trace, e.g. RT × CL, RT × CS or RT × CL&CS. We implemented the new models by changing Eq. (13) in the following ways (note that we explicitly write down the multiplication symbol as × for clarity):

$$\text{DV}_{\text{RT}\times\text{CL},t} = \text{DV}_{\text{RLsimple},t} + w_{\text{CL}} \times \text{R-trace}_t \times \text{CL-bonus}_t + w_{\text{CS}}\text{CS-bonus}_t \tag{23}$$

$$\text{DV}_{\text{RT}\times\text{CS},t} = \text{DV}_{\text{RLsimple},t} + w_{\text{CL}}\text{CL-bonus}_t + w_{\text{CS}} \times \text{R-trace}_t \times \text{CS-bonus}_t \tag{24}$$

$$\begin{aligned}\text{DV}_{\text{RT}\times\text{CL\&CS},t} = \text{DV}_{\text{RLsimple},t} &+ w_{\text{CL}} \times \text{R-trace}_t \times \text{CL-bonus}_t \\ &+ w_{\text{CS}} \times \text{R-trace}_t \times \text{CS-bonus}_t\end{aligned} \tag{25}$$

These models mediate the impact of R-trace on the decision variable via the choice memory traces. These models do in fact have the advantage compared to the full model that they use one less free parameter ($w_{\text{R}}$). However, note also that in these models, the impact of choice memory traces is entirely dominated by R-trace, meaning that if R-trace is zero, any tendency to repeat choices is abolished. This

might not reflect the way in which R-trace modulates choice traces; instead, it might be more plausible to assume a constant effect of choice repetition which is then to some degree upscaled or downscaled by R-trace. For this reason, we constructed additional similar control models that assume an additive effect of R-trace on the choice memory trace weight via an additional free parameter. We denote these models with RT × CL 2, RT × CS 2 or RT × CL&CS 2:

$$DV_{RT \times CL\,2,t} = DV_{RLsimple,t} + (w_{CL} + w_{RT \times CL} \times R\text{-trace}_t) \times CL\text{-bonus}_t \\ + w_{CS}CS\text{-bonus}_t \tag{26}$$

$$DV_{RT \times CS\,2,t} = DV_{RLsimple,t} + w_{CL}CL\text{-bonus}_t + (w_{CS} + w_{RT \times CS} \times R\text{-trace}_t) \\ \times CS\text{-bonus}_t \tag{27}$$

$$DV_{RT \times CL\&CS\,2,t} = DV_{RLsimple,t} + (w_{CL} + w_{RT \times CL} \times R\text{-trace}_t) \times CL\text{-bonus}_t \\ + (w_{CS} + w_{RT \times CS} \times R\text{-trace}_t) \times CS\text{-bonus}_t \tag{28}$$

In close analogy to these control models assessing a mediation of R-trace effects via choice memory traces in a multiplicative or additive fashion, we also considered whether R-trace acts by modulating the inverse temperature parameter from the softmax function (Eq. (4)). We constructed two new models, again without an influence of R-trace on the PE calculation. Instead, R-trace modulated the β parameter of these models. The new β parameters for these models were constructed as:

$$\beta_{RT \times \beta,t} = \beta \times R\text{-trace}_t \tag{29}$$

$$\beta_{RT \times \beta\,2,t} = \beta + w_{RT \times \beta} \times R\text{-trace}_t \tag{30}$$

All of the above models belong to the first category of supplementary control models we examined. They all have in common that they assume an effect of R-trace on choice. However, they all assume that this effect is mediated via different routes compared to the one we use in the full model, i.e. instead of influencing value update in the PE, they assume that R-trace acts on non-value-related mechanisms of the model.

By contrast, the second category of control models that we examined does not require a reward trace at all. As explained in the main text, the full model effectively implements asymmetric (i.e. different for positive and negative outcomes) and dynamic (depending on whether R-trace is low or high) value updates. The first control model from this category therefore examines whether a model with no reward trace, but instead asymmetric learning rates[41] fits the data better than the full model. This model (AsyAlpha) makes use of the architecture of the full model without R-trace. This also removed the weight parameter, $w_R$, of R-trace, as well as the R-trace specific learning rate $\alpha_R$. Instead, the AsyAlpha model uses separate learning rates for rewarded and for unrewarded trials. The final control model is also similarly based on the architecture of the full model without R-trace but assumes a dynamic learning rate (DynAlpha). It is based on the widely held idea that learning rates should increase when surprising outcomes are encountered, whereas it should decrease if outcomes conform to expectations. One RL implementation of this idea tracks the slope of absolute prediction errors to upregulate or downregulate the learning rate[26,62]. Note that this model is not, as the others, nested within more basic models. It uses the free parameter $\alpha_R$ as the initial learning rate on the first trial, but also to control the smoothing when tracking the absolute prediction errors (PEmag):

$$PEmag_t = (1 - \alpha_R)PEmag_{t-1} + \alpha_R|PE_t| \tag{31}$$

From this, the normalized slope of absolute prediction errors ($m$) is calculated:

$$m_t = \frac{PEmag_t - PEmag_{t-1}}{(PEmag_t + PEmag_{t-1})/2} \tag{32}$$

The slope is then used to generate a link function using a free parameter γ:

$$fm_t = sign(m_t)\left(1 - e^{-\left(\frac{m_t}{\gamma}\right)^2}\right) \tag{33}$$

And this link function is then used to calculate the learning rate (LR) for the current trial:

$$LR_t = \begin{cases} LR_{t-1} + fm_t \times (1 - LR_{t-1}) & \text{if } m_t \geq 0 \\ LR_{t-1} + fm_t \times LR_{t-1} & \text{if } m_t < 0 \end{cases} \tag{34}$$

**Effective learning rate calculation**. As noted in the main text, the effect of R-trace in the full model RL + cl + cs + rt is that it impacts the PE calculation. This means that it increases or decreases the PE based on the current GRS, which will increase or decrease the change in Q-value. Such a mechanism can produce asymmetric and dynamic changes in value updates without explicitly changing the learning rate. Our model predicts that value updates from the same outcome event are higher for rewarded trials when R-trace is high compared to low and vice versa for unrewarded trials. We specifically tested this hypothesis by calculating the effective learning rates from our full model. We did this by running an additional model analysis. For this analysis, we fitted a new model that did not contain an explicit R-trace. In addition, we fixed the free parameters for each session to the session-specific parameter values from the full model. The model comprised four free

parameters, all of which were learning rates, but different ones for rewarded and unrewarded trials, and ones with a high (i.e. ≥0.5) or low (i.e. <0.5) R-trace. Note that the R-trace information was session-specific and imported from the full model; it was not calculated or fitted in effAlpha itself. In other words, what this model did was to assume that different effective learning rates were used for positive and negative outcomes depending on high or low R-trace. It keeps all other features identical to the full model by fixing the remaining parameters to the ones from the full model. But instead of using R-trace in the PE calculation, it examines whether the effective learning rates indeed differ in the manner expected by our full model. Note that in order to test the learning rates from effAlpha for significant differences, we fitted the model session-wise via maximum likelihood estimation, because the hierarchical fit decreases the variance of individual parameters which possibly biases comparisons of free parameters.

**Model simulations**. We ran model simulations to illustrate how an RL model including reward traces can effectively lead to the assignment of rewards to choices that occur after (rather than before) that reward was obtained. We simulated sequences of choices for every session of our three experiments using the true underlying reward schedule. Then, we concatenated sessions from the same monkey within each experiment (just as had been done in the behavioral analysis in Fig. 1) and applied a behavioral GLM to the simulated choices. This behavioral GLM is explained below and is different from the behavioral GLM in Fig. 1. The resulting 12 (3 experiments with 4 monkeys) beta weights per regressor were averaged and are shown in Fig. 4.

To isolate the effects of R-trace on decision making in our simulations, we removed effects related to choice repetition and side bias by setting $\lambda_{CS}$, $w_{CS}$, $\alpha_{CL}$, $w_{CL}$ and SB to zero. Moreover, we set the remaining model parameters to standard values ($\alpha = 0.35$, $\beta = 1$, $\alpha_R = 0.5$) and systematically varied the effect of R-trace in the prediction error calculation (see Eq. (14)). For this, in separate iterations of the simulation, we set $w_R$ to −1, 0, 1, and 2. This determines whether R-trace has a negative effect, no effect, or a positive effect in the prediction error calculation. We derived choice probabilities from the simulated Q-values by using the softmax equation above (Eq. (4)) and used these choice probabilities to generate simulated choices.

The logistic GLM we applied to the simulated choices was similar to previous studies[11,13]. The GLM considered the last 5 choices and the last 5 outcomes. For each of the three choice stimuli, we applied a separate GLM and combined their beta weights to a covariance-weighted mean. We used 25 regressors in the GLM; one for each choice-outcome conjunction. The conjunctions were coded as 1, −1 or 0 depending on whether the choice of interest was made and rewarded, another choice was made and rewarded or no reward occurred, respectively. As in previous reports, regressors relating to the last time points of choices or rewards (e.g. Choice(t-5) and reward(t-1)) were confound regressors and are therefore not shown in Fig. 4.

**Imaging data acquisition**. Awake-animals were head-fixed in a sphinx position in an MRI-compatible chair. We collected fMRI using a 3T MRI scanner and a four-channel phased array receive coil in conjunction with a radial transmission coil (Windmiller Kolster Scientific Fresno, CA). FMRI data were acquired using a gradient-echo T2* echo planar imaging (EPI) sequence with $1.5 \times 1.5 \times 1.5\,\text{mm}^3$ resolution, repetition time (TR) = 2.28 s, Echo Time (TE) = 30 ms, flip angle = 90, and reference images for artifact corrections were also collected. Proton-density-weighted images using a gradient-refocused echo (GRE) sequence (TR = 10 ms, TE = 2.52 ms, flip angle = 25) were acquired as reference for body motion artifact correction. T1-weighted MP-RAGE images ($0.5 \times 0.5 \times 0.5\,\text{mm}^3$ resolution, TR = 2,5 ms, TE = 4.01 ms) were acquired in separate anesthetized scanning sessions.

**fMRI data preprocessing**. FMRI data were corrected for body motion artifacts by an offline-SENSE reconstruction method[63] (Offline_SENSE GUI, Windmiller Kolster Scientific, Fresno, CA). The images were aligned to an EPI reference image slice-by-slice to account for body motion and then aligned to each animal's structural volume to account for static field distortion[64] (Align_EPI GUI and Align_Anatomy GUI, Windmiller Kolster Scientific, Fresno, CA). The aligned data were processed with high-pass temporal filtering (3-dB cutoff of 100 s) and Gaussian spatial smoothing (full-width half maximum of 3 mm). The data that were already registered to each subject's structural space were then registered to the CARET macaque F99 template[65] using affine transformation.

**fMRI whole-brain analysis**. We employed a univariate approach within the general linear model (GLM) framework to perform whole-brain statistical analyses of functional data as implemented in the FMRIB Software Library version 5.0.11[66] where each of the psychological regressors was convolved with a hemodynamic response function (HRF) specific for monkey brains[67,68]. Using this framework we initially performed a first-level fixed effects analysis to process each individual experimental run. These were then combined in a second-level mixed-effects analysis (FLAME 1) treating session as a random effect (we also had a similar number of sessions across subjects) and using family-wise error cluster correction ($z > 2.6$ and $p = 0.05$). One reason for omitting an intermediate level of MRI analysis was that the subject number in our study, as in the majority of studies of macaque neural activity, was below the one required for random effects analyses across individual monkeys[69].

Our first-level GLMs had several features in common. When using regressors derived from the modeling, they were always taken from the full model $RL + cl + cs + rt$ and were estimated session-wise based on the mean of the Gaussian parameter distributions. All GLMs employed the same basic set of regressors to capture variance in the data. A decision constant (DEC) was time-locked to the onset of the decision screen and the duration was set to the reaction time for that trial. We used two constant regressors time-locked to the time of feedback, one for rewarded outcomes (FBrew) and one for unrewarded outcomes (FBnorew). Both had a default duration of 0.1 s. All subsequent regressors in all GLMs were set to this default duration unless otherwise noted. In addition, to account for movement related scanning artifacts, we used unconvolved regressors for leftwards and rightwards responses (Cleft_unc and Cright_unc), time-locked to the respective responses, with the duration set to the TR (2.28 s). Except for Cleft_unc and Cright_unc, all regressors in all GLMs were convolved with the monkey-specific HRF (mean lag of 3 s, standard deviation of 1.5 s). Finally, we used a binary regressor to indicate choice location (1 for right, 2 for left), time-locked to choice (Cloc). As all other subsequent binary or parametric regressors, Cloc was z-scored before entering it in the GLM.

Our first GLM focused on the memory traces specific to our RL model (GLM1). Time-locked to DEC, it contained the Q-value difference between the chosen and the unchosen option. This quantity reflects the value driven part of the decision variable and we refer to it as $DV_{value}$(chosen-unchosen).

$$DV_{value}(chosen - unchosen) = Q(chosen) - Q(unchosen) \quad (35)$$

In addition, the GLM contained the Q-value of the unpresented option (Q-unp), the CL-trace (CL-trace), the CS-trace of the unpresented option (CS-unp), as well as the CS-trace comparison between the chosen and the unchosen option (CS(chosen-unchosen)). Moreover, we included the R-trace in the GLM (R-trace), which was time-locked to the onset of the feedback screen, as in the model, the reward trace affects the prediction error at the time of learning. These additional regressors were z-scored and had a default duration of 0.1 s. Moreover, we calculated the contrast between rewarded and unrewarded outcomes (FBrew-FBnorew) and refer to this contrast as the outcome effect. For the second level analysis of the outcome contrast specifically, we employed a restrictive whole-brain pre-threshold mask to exclude movement artifacts related to the consumption of the juice rewards. However, the reported effects remain significant even without this mask.

In our second GLM (GLM2), we used the same set of regressors as in GLM1 with one difference. We split up the R-trace regressor in rewarded and unrewarded trials. This means we time-locked this effect to only rewarded and only unrewarded trials (R-trace (R) and R-trace (no R)), respectively, and the parametric variation of R-trace was z-scored within the respective trial set.

Our third GLM (GLM3) investigated decision related activity by regressing the BOLD signal on the decision variable of the RL model in addition to the set of basic regressors described above. No further regressors were included in the GLM. We coded it in terms of chosen minus unchosen, i.e. as relative evidence for the chosen option (DV(chosen-unchosen)). It was time-locked to DEC onset. For brevity, DV(chosen-unchosen) is just referred to as $DV_{total}$ in the main text.

The final whole-brain GLM (GLM4) broke apart the decision variable in one element related to value and one related to the previous choice history, $DV_{value}$(chosen-unchosen) (same as in GLM1 and GLM2) and $DV_{choice}$(chosen-unchosen) (see Fig. 2d). These two regressors were the only ones added to the basic set of regressors and they were coded in terms of chosen minus unchosen, similar to the decision variable in GLM 1. Specifically, the two quantities were defined as:

$$\begin{aligned} DV_{choice}(chosen - unchosen) = DV(chosen - unchosen) \\ - DV_{value}(chosen - unchosen) \end{aligned} \quad (36)$$

$DV_{value}$(chosen-unchosen) and $DV_{choice}$(chosen-unchosen) were both time-locked to DEC. For brevity, $DV_{value}$(chosen-unchosen) and $DV_{choice}$(chosen-unchosen) are just referred to as $DV_{value}$ and $DV_{choice}$ in the main text.

**Region of interest analyses**. We analyzed ROI data using Matlab 2018a version 9.4.0, Jasp version 0.9.0.1, and SPM version 12. For all ROI analyses, we used spherical ROIs with 2 mm radius centered on peak coordinates reported in Supplementary Table 1. First, to examine R-trace related effects during trials in which either reward was or was not delivered, we read out FSL's contrast of parameter estimate (COPE) maps in independently selected ROIs by warping our spherical mask in session-specific space and averaging the parameter estimates over the mask. To account for outlying data points, we square-root transformed the data. In case of negative COPE values, we used the absolute value and added a minus after the transformation. For ROI analyses of COPE effects where the ROI focused on a similar effect to the defining whole-brain contrast, ROIs were selected via a leave-one-out method to avoid selection bias. For this, we calculated the whole-brain contrast of interest over all sessions but kept one session out at a time. ROI peak coordinates for the contrast were then identified based on the (incomplete) group average (that did not include the left-out session) and applied to the left-out session. We repeated this procedure for all sessions and applied statistical tests to the resulting independent ROIs.

We used the ROI timecourse analyses to examine the temporal evolution of neural signals in our ROIs. To this end, we extracted the preprocessed BOLD time series and

up-sampled it by a factor of 10 (using spline intrapolation) and aligned the time series trial-wise. On every time point, we applied a set of z-scored regressors to the data. The time course GLM (tGLM1) performed the following calculations to each time point of the upsampled data in a session by session manner. First, we used a regressor set comprising $DV_{value}$(chosen-unchosen), $DV_{choice}$(chosen-unchosen), Q-unp, CS-unp, CL-trace, and Cloc. After applying this regressor set to the data time-locked to the feedback phase onset with a linear GLM, we calculated the residuals of the BOLD timecourse. Then, we binned the BOLD residuals by outcome (rewarded or unrewarded) and by R-trace (high or low; median split). On the group level we performed two statistical procedures with the binned BOLD residuals. First, we applied a $2 \times 2$ analysis of variance (ANOVA) to it on every time point and calculated main effects of outcome and R-trace as well as the interaction. Second, for each session, we calculated the peak times of relevant ANOVA effects to be able to compare when effects were strongest both within and across brain regions. As dACC represented R-trace preferentially during rewarded trials and DRN preferentially represented R-trace during no reward trials, we calculated separate measures of the R-trace effect for rewarded and unrewarded trials for dACC, DRN, and Ia. For every time step and session, we subtracted the residual BOLD time course of R-trace low from R-trace high separately for rewarded and unrewarded trials, resulting in the effects R-trace (R) and R-trace (no R), respectively. Moreover, we calculated the overall difference of rewarded minus unrewarded bins to obtain the effect of outcome. Within a plausible time window of two to six seconds (given the relatively fast macaque hemodynamic response function[19]) from feedback onset, we determined the session-specific time of the peak of the relevant effect. As we are not testing the size of the peak effect but only the time for each session at which it occurs, restricting the effects to a positive or negative direction does not bias the results. For each effect, we constrained the analysis to identify the timing of effect peaks in the direction concordant with the group effect. This means for dACC, we looked for the time of the most negative R-trace effect during rewarded trials as well as the most negative outcome effect. For Ia bilaterally, we looked for the time of the most positive peak for each effect as both effects were positive. For the DRN, we took the time of the most positive outcome effect and the time of the most negative R-trace effect during unrewarded trials.

**Neural model comparison**. To assess whether vmPFC signaled a combined decision variable rather than value difference alone, we performed a neural model comparison[17,20]. Based on the $DV_{total}$ signal from GLM3, we used a leave-one-out procedure to select session-specific vmPFC ROIs and avoid biased selection. Then, we first regressed the vmPFC BOLD signal against GLM3. Second, we did the same for GLM4, but crucially without the inclusion of $DV_{choice}$. This means that both GLMs contained identical regressors, with the exception that the first one contained $DV_{total}$ and the second one $DV_{value}$ instead. We used Matlab's fitglm function together with the convolved design matrix and the BOLD timecourse for this regresssion and extracted the maximum log likelihood estimates from the resulting model. These log likelihood estimates from the two GLMs were then fed into a Bayesian model selection random-effects analysis (using the spm_BMS routine from SPM12[29]), which computed the exceedance probability of each GLM. This analysis indicated which GLM best explained the neural data.

**Reporting summary**. Further information on research design is available in the Nature Research Reporting Summary linked to this article.

## Data availability

We have deposited all choice raw data in an OSF repository. All reinforcement learning results in this paper are derived from these data alone. The accession code is: https://osf.io/358cg/?view_only=0e6fda7925364d86930374cd4ae4a59f. Any remaining data that support the findings of this study are available from the corresponding author upon reasonable request. We have also deposited all group-level contrast images presented in the manuscript on Neurovault. The accession code is: https://neurovault.org/collections/KJVDIJYY/. The source data underlying Figs. 1f–g, 2e–i, 3c–g, 5, 6c, d, 8b, and 9c and Supplementary Figs. 1b, c, 2i–l, 3a–r, 4a–k, 5c, f, and 7 are provided as a Source Data file.

## Code availability

The above repository also comprises the full reinforcement modeling pipeline including model comparisons implemented in Matlab. All variables used for the MRI analyses are derived from this pipeline. In addition, code for the primary behavioral GLM is included (Fig. 1f, g). Accession code to the repository is the following and a README inside the repository explains the details of its use: https://osf.io/358cg/?view_only=0e6fda7925364d86930374cd4ae4a59f. Any remaining code that support the findings of this study are available from the corresponding author upon reasonable request.

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

## Acknowledgements

This study was funded by the Wellcome Trust (WT100973AI; 203139/Z/16/Z; 105238/Z/14/Z) and MRC (MR/P024955/1; CQR01330) and a UKRI FLF to E.F. (MR/T023007/1). M.K. was funded by the Centre National de la Recherche Scientifique (Mission pour l'Interdisciplinarité). We thank Tim Behrens, Nils Kolling, Alizée Lopez-Persem, and Jacqueline Scholl for helpful discussions.

## Author contributions

E.F., D.F., B.K.H.C., and M.F.S.R. designed the experiments. D.F. and B.K.H.C. collected the data. M.K.W., E.F., and M.K. carried out data analyses. M.K.W., E.F., D.F., M.K., M.C.K.F. and M.F.S.R. interpreted the results. M.K.W. and M.F.S.R. drafted the manuscript and all authors commented on the manuscript.

## Competing interests

The authors declare no competing interests.
