## [Peer Review File · Nature Communications]

Reviewers' Comments:

Reviewer #1:

Remarks to the Author:

This paper shows that the average reward rate during associative learning (global reward state, GRS) affects choice. Four macaques participated in two behavioral and one fMRI experiments. All experiments consisted of trials in which the monkey chose between two cues; correct choices led to juice reward. Monkeys exhibited win-stay-lose-switch tendencies, but these tendencies were affected by GRS. Successful choices were repeated more in rich, compared to poor, environments, while switches from unsuccessful choices are more likely in poor environments. Behavior is described with a reinforcement learning model that includes the reward trace, among other parameters. Activation in anterior insula and dorsal raphe nucleus reflects both immediate outcome and GRS, whereas activity in ventromedial prefrontal cortex reflects the difference between option values.

This is a very interesting and clearly written paper. The topic is important, the combination of behavioral modeling and fMRI in monkeys is informative, and the analysis, including behavioral model comparisons and fmri analysis, is careful and methodical. However, some clarifications are needed, as detailed below.

- It wasn't clear to me how data from the four monkeys were handled in the behavioral regression. Was this a fixed-effects analysis or were the individuals accounted for? The main text refers to the online methods, but I couldn't find a description of this analysis there. Could you show similar regressions within monkey, or at least divide the data into two and show replication in the two halves?

- A similar question regarding the model-fitting. The text mentions hierarchical fitting over sessions, but I wasn't sure exactly how sessions and monkeys were treated. Also, it is encouraging that in 47 out of 65 sessions, the best model outdid the second-best model, but could you clarify whether these sessions were spread across all four monkeys? Was the best model favored in the majority of sessions within each monkey?

- The authors do a thorough job in comparing models with all the possible permutations of CL, CS and R-trace. It wasn't clear to me, however, that the effect of GRS could only be exerted as influence on the PE. For example, perhaps motivation to switch is lower when GRS is higher (lower need for exploration)? In other words, is it possible that GRS affects the weighting of CL-trace and/or CS-trace? Also, the authors stress that their model can reproduce dynamic changes in value update rates and asymmetries in value updates without explicitly adjusting learning rates. Is it possible to explicitly compare with a model that does adjust learning rates? Can you explain why one is more likely than the other?

- Similar to the behavioral analysis, in the imaging analysis it was also not clear to me how the four individuals were accounted for. The methods section mentions that session was treated as a random effect – but there were multiple sessions per monkey, doesn't that inflate the significance of the results? Can you show results within monkey?

- The authors report activation patterns that are consistent with their model, but also with other models. They do not search for encoding of PE, which could be very helpful here. If activity in areas like the ventral striatum, where PE is usually observed, is indeed modulated by R-trace, it will support the particular model the authors put forth.

- The authors mention that in other contexts they and others have observed contrasting, rather than facilitating, effects of GRS. Could you discuss the different contexts, when is GRS weighted positively

and when negatively? How do the results relate (or not) to previous work about menu normalization effects?

Reviewer #2:

Remarks to the Author:

This paper proposed a new RL model in which the calculation of reward prediction error is biased by a number tracking the recent average reward rate (global reward state). This bias causes the RL agent to have a higher chance of repeating the actions being rewarded in a time period when the agent has often been rewarded regardless of the choices (rich context), and a higher chance to avoid unsuccessful choice in a poor context. In addition to explaining monkey behavioral choice, the paper also identified anterior insular and dorsal raphe nucleus as regions that track this global reward state. In contrast, neural signal in vmPFC only reflects the value difference between the two choices, but not the global reward state.

To my knowledge, the idea of a global reward state is quite novel, and the implication is intriguing: instead of assigning credit of current reward to some choices made in the past, monkeys appear to in fact assign credit of past reward to current choice. This is quite a surprising and interesting finding.

The experiment design is beautiful. The analysis procedure appears generally solid, except for a few confusions in the method (see below). The paper is well written. Graph illustrations are clear. Major codes and behavioral data are shared. I suggest the authors considering sharing the fMRI data as well.

I generally recommend acceptance, but I think minor revision to address the points below would enhance the paper:

1. Online Method, Page 27, EM procedure. Based on the description, it feels to me the procedure is actually coordinate descent instead of EM. I don't see a clear marginalization procedure typical in the expectation step of EM. Equations 15 and 16 are inconsistent with the paragraph above. In the equations, there is a sum of prior probability over μ and σ , but the paragraph seems to suggest only a single value of μ and σ are used. Indeed, Equations 17 and 18 specify how point estimates of μ and σ are calculated. And to me, it also makes no sense to sum the log prior probability in the second term of equation 15 and 16.

The acronym NPL seems to have no relation with the term "maximum posterior likelihood". Also, I suggest using "posterior probability" instead of "posterior likelihood" (think of prior, likelihood and posterior in Bayesian framework: likelihood has a specific meaning and it is better not to use it to replace "probability" in a general sense).

To be more accurate, I also suggest using $p(\text{choice}_t | h, \text{choice}_{\{1, \dots, t-1\}})$ in place of $p_t(\text{choice} | h)$. Strictly speaking, the likelihood of each trial is not independent, only the conditional probability of the current trial given the DV (which depends on all the past choices, past stimuli and h) is independent across trials and thus can be multiplied.

It may be better using Latin letters μ and σ instead of the words μ and σ .

2. Is the Hessian matrix calculated based on just the likelihood (which does not include prior), or the posterior? If posterior, is it based on the initial prior (μ and σ), or the recalculated μ and σ ? I feel that the Hessian of the likelihood or the posterior using the initial prior may be the correct way, at least for the purpose of Equation 22. Please state it clearly.

3. Page 36, in neural model comparison, it is worth specifying how the log-likelihood used for Bayesian model selection is calculated. Is it marginal likelihood, or maximal likelihood?
4. Figure 1g. I did not find a description of how the residual probability is calculated.
5. Fig 5e, f: the ANOVA over so many time points without mentioning a control for multiple comparisons is worrying. Also, due to the upsampling, adjacent time points would be highly correlated and redundant. So the F-statistics and p-value should really be just for illustrative purposes and not for drawing conclusions. The lack of control for multiple comparisons should be clearly stated.
6. The caption for supplementary figure 1a says the result does not change with history length. I think it is worth specifying what range of history length has been tested.
7. Figure 2 had dashed line without explanation (it only appeared in the caption for Fig 7)
8. On Page 13 the term "residual BOLD" suddenly appears without any explanation of what it is. Although the procedure is explained in methods, it is worth briefly explaining what it is to help the reader understand.
9. There are multiple places where the authors described regressing some model parameters (DV, GRS etc.) against BOLD signal. My understanding is that when we say "regressing A against B", we treat A as a dependent variable and B as an independent variable. So it seems the phrasing should be regressing BOLD signals against these regressors instead of the other way.
10. The bottom two lines in 14: I think what the authors wanted to write may be that DRN and dACC showed earlier outcome encoding compared to Ia (not Ia or dACC)
11. Is the posterior density of w_f in Fig 3a over all sessions (i.e., based on the final μ and σ)? It may be worth explaining it somewhere.

Reviewer #3:

Remarks to the Author:

and reversal paradigm to show that macaques not only use choice-outcome associations to guide decision making, but also uses the recent overall reward state of the environment (Global reward state; GRS) to do alter those choice-outcome valuations. When GRS is high (lots of rewards around) successful choices are more likely to be repeated, but when GRS is low, loss-associated choices are less likely to be repeated. Modelling of this process suggests that this is due the GRS biasing the prediction errors associated with the task feedback. MRI scanning of task performance and integration with this model indicates important roles for the anterior insula in integrating new outcomes with the overall GRS, and for the dorsal raphe nucleus in the relative weighting of outcomes.

In general I think it is a very nice study that is well written and may also provide explanations for other effects (as the authors consider in their discussion). My comments are therefore relatively minor:

- 1) How is the GRS defined for the anova analysis? Is it the last 5 trials? If so this needs to be explicit in the text as I only eventually found it in a figure legend which suggests that the "last 5 trials" were

used in general for the GLMs. Why was 5 trials chosen rather than say 3 or 8?

2) Page 30 deals with the GLM applied to simulated choices, and I'm not clear if this is identical to the one used for real choices (p6)?

3) I have missed the definitions of how CxR-history, C-history, and R-history (as per p6) were calculated?

4) Why was the influence of the R trace only investigated for prediction errors? For example, it could also have been used to influence the softmax temperature of reward-based decision-making?

Reviewers' comments:

Reviewer #1 (Remarks to the Author):

This paper shows that the average reward rate during associative learning (global reward state, GRS) affects choice. Four macaques participated in two behavioral and one fMRI experiments. All experiments consisted of trials in which the monkey chose between two cues; correct choices led to juice reward. Monkeys exhibited win-stay-lose-switch tendencies, but these tendencies were affected by GRS. Successful choices were repeated more in rich, compared to poor, environments, while switches from unsuccessful choices are more likely in poor environments. Behavior is described with a reinforcement learning model that includes the reward trace, among other parameters. Activation in anterior insula and dorsal raphe nucleus reflects both immediate outcome and GRS, whereas activity in ventromedial prefrontal cortex reflects the difference between option values.

This is a very interesting and clearly written paper. The topic is important, the combination of behavioral modeling and fMRI in monkeys is informative, and the analysis, including behavioral model comparisons and fmri analysis, is careful and methodical. However, some clarifications are needed, as detailed below.

We thank the reviewer for the careful review and the positive evaluation. We clarify the highlighted points below and found that they indeed had a very positive impact on the presentation and clarity of our analyses and results.

- It wasn't clear to me how data from the four monkeys were handled in the behavioral regression. Was this a fixed-effects analysis or were the individuals accounted for? The main text refers to the online methods, but I couldn't find a description of this analysis there. Could you show similar regressions within monkey, or at least divide the data into two and show replication in the two halves?

The behavioral regression did in fact account for the identity of the individual monkeys. We apologize that the online methods indeed did not further specify this. We have changed this in our revised manuscript. In our regression analyses we concatenated experimental sessions per animal per experiment and applied the GLM to the concatenated sessions. As all four animals participated in all three experiments, this procedure resulted in 12 beta weights (per regressor) overall, three per animal. This meant that the sessions from the same monkey within an experiment were treated as fixed effects, but not the monkeys themselves. The reason for this procedure was that our main behavioral GLM used a very extensive regressor set (22 regressors overall) and to be able to estimate reliable effect sizes, we needed to concatenate the choice data.

We now explain this procedure in our Methods section as follows:

Behavioral data analysis. We ran logistic general linear model (GLM) analyses, implemented in Matlab 2018a version 9.4.0, on the behavioral choice data. To be able to estimate complex GLMs with a large regressor set, we concatenated experimental sessions per animal per experiment and applied the GLM to the concatenated sessions (Fig.1f,g; Supplementary Fig.1b,c; Fig.3I uses the same procedure with simulated choice data). Note that Supplementary Fig.1 describes all 22 regressors used in our main behavioral GLM. As all four animals participated in all three experiments, this procedure resulted in 12 beta weights (per regressor) overall, three per animal. We tested these resulting beta weights for significance using analyses of variance (ANOVA) and one-sample t-tests against zero, implemented in Matlab and Jasp version 0.9.0.1. For GLMs with smaller regressor sets effect sizes were estimable on the basis of single sessions. In such a case we applied the GLMs separately for each of the 65 sessions collapsed over experiments, for instance for the reaction time

GLMs. Reaction time analyses used a linear link function. Beta weights were tested for significance using one-sample t-tests against zero.

In addition to clarifying our behavioral GLM, we now indicate the identity of the individual monkeys in all relevant figures in our manuscript, not only in the figures relating to the behavioral regression. We use the same symbols to indicate the individuals in all these figures. This has led us to change over 20 figure panels in our manuscript. To keep this response letter readable, we show below just the revised figure of our main behavioral regression reported in the main text together with the clarified part of the figure legend:

Figure 1. Task design and behavior. ... panels f,g concatenate sessions per monkey per experiment resulting in three data points per individual; error bars are SEM across monkey data sets; $n = 12$; *, $p < 0.05$). Source data are provided as Source Data file. Symbols indicate monkey identity in panels f,g; “MK” abbreviates “monkey”.

Finally, as we have explained above, it is very difficult to estimate our behavioral GLMs on the level of individual monkeys because of the large number of regressors, and it was this consideration that led us to concatenate the data in the first place. We hope we have made it clear that our regression preserves the identity of the individuals and we now show this very clearly in all relevant figures by using distinct symbols for each individual. Nevertheless, we agree that it would be very intriguing to be able to look at these effects at the level of individual monkeys and so we have looked into possibilities for doing this. As a result, we adopted a GLM approach using ridge regression and are now also able to demonstrate our regression effects in individual monkeys. In three out of four monkeys, we found a very significant effect of our key variable, R-history. This clearly demonstrates that the effect of the global reward state on choice behavior is not driven by an outlying individual. We implemented the ridge regression analyses in the following way:

Unlike in our main analysis in which we concatenated sessions, the same regression model was now applied to all individual sessions. Ridge regression penalizes large beta weights according to a regularization coefficient λ and thus prevents overfitting and improves generalization. This is appropriate for cases such as ours when there are many regressors and comparatively few trials (200 trials per session). We applied the regression model to all sessions using Matlab’s *lassoglm* (setting Alpha to a very small value in the following way. First, we determined an appropriate regularization coefficient λ for each individual macaque. To do this we applied a five-fold cross-validation to the data sampling over a large space of λ from zero to 10^{-4} to 10^4 (log spaced) to each individual session. We repeated this procedure 10 times and determined the overall model deviance for each λ for all sessions from the same monkey combined. We then selected the λ that resulted in the lowest overall model deviance for each monkey indicating the best cross-validated model fit. A single λ was used for all sessions (irrespective of experiment) from the same monkey to keep regression weights to scale. The selected λ s reflect the best trade-off of model parsimony and predictive power. After selecting macaque-specific λ s in this way, we applied the regression model using these λ s. Then, again, we tested our critical effect of interest, the R-history effect, in all monkeys to determine whether we were able to find evidence for it on the individual monkey level. Indeed, in three out of four monkeys, we found a significant effect of R-history using one sample t-tests.

We added a new supplementary figure to describe these analyses and show the results:

Supplementary Figure 2. Behavioral GLM analysis of individual macaques using ridge regression. To apply our regression model (shown in Fig.1f) to individual macaque data, we used ridge regression¹. Unlike in our

main analysis in which we concatenated sessions, the same regression model was now applied to individual sessions. Ridge regression penalizes large beta weights according to a regularization coefficient λ and thus prevents overfitting and improves generalization. This is appropriate for cases such as ours when there are many regressors and comparatively few trials. We applied the regression model to all sessions using Matlab's *lassoglm* (setting Alpha to a very small value) in the following way. First, we determined an appropriate regularization coefficient λ for each individual macaque. To do this we applied a five-fold cross-validation to the data sampling over a large space of λ from zero to 10^{-4} to 10^4 (log spaced) to each individual session. We repeated this procedure 10 times and determined the overall model deviance for each λ for all sessions from the same monkey (irrespective of experiment) combined. We then selected the λ for each individual macaque that resulted in the lowest overall model deviance for that monkey; this is the λ for the best cross-validated model fit. A single λ was used for all sessions from the same monkey to keep regression weights to scale. **(a-d)** Deviance averaged across sessions plotted over λ in log space. The red dot indicates the point of lowest deviance (i.e. best cross-validated model fit), which was used to pick the monkey specific λ . The resulting λ s for the four monkeys were close together: 0.17074 (monkey 1), 0.24771 (monkey 2) 0.11768 (monkey 3), 0.24771 (monkey 4) and reflect the best trade-off of model parsimony and predictive power. **(e-f)** All beta weights (except the model intercept) averaged over sessions and plotted over log spaced λ . The vertical red line indicates the λ used for the monkey. Note that beta weights decrease along the λ axis as the penalty for large beta weights increases. Panels in the same column correspond to the same individual **(i-l)** After selection of macaque-specific λ s in such a way, we applied the regression model using these λ s. Then, again, we tested our critical effect of interest, the R-history effect, in all monkeys to determine whether we were able to find evidence for it on the individual monkey level. Indeed, we found a significant effect of R-history in three out of four monkeys using one sample t-tests. (monkey 1, n=16: $t_{15} = 4.210$; $p < 0.001$; monkey 2, n=17: $t_{16} = 5.201$; $p < 0.001$; monkey 3, n=15: $t_{14} = 0.425$; $p = 0.677$; monkey 4, n=17: $t_{16} = 6.426$; $p < 0.001$). Error bars are SEM across sessions; *, $p < 0.05$). Source data are provided as Source Data file. Symbols in panels l,j,k,l indicate monkey identity according to the same convention as in all other figures.

- A similar question regarding the model-fitting. The text mentions hierarchical fitting over sessions, but I wasn't sure exactly how sessions and monkeys were treated. Also, it is encouraging that in 47 out of 65 sessions, the best model outdid the second-best model, but could you clarify whether these sessions were spread across all four monkeys? Was the best model favored in the majority of sessions within each monkey?

We can see that this question aims at a similar issue as the question before and so we also addressed it in a similar fashion. We think this is a particularly direct way to address the concerns, but also one that will make it easy for the reader of the manuscript to follow the line of argument. First, we clarified our fitting procedure in the Methods. Second, we changed the figure the reviewer is referring to that shows the advantage of the best-fitting model over the second-best one; it now also indicates monkey identity, and this makes it possible to compare the strength of the effects between individuals. Finally, we now conduct our model comparison also for each monkey individually. The latter approach is in our view the most direct and comprehensive way of answering the question whether our model is the most appropriate description of each monkey's choices. Indeed, paralleling the results from the behavioral ridge regression approach, we found that our model using a choice-unrelated reward trace clearly wins the model comparison in the same three out of the four monkeys in which we could also find behavioral evidence for a modulation of choice by the general reward state. For the remaining monkey, there was no clear winning model as the evidence for different models was too close together for an unambiguous decision.

First, we clarify the fitting approach. In all cases the fitting comprised two levels: the lower level of the individual sessions and a higher-level. In our main modelling analysis, the higher level is the one for all sessions together irrespective of monkey and experiment. This approach enables us to estimate a group-level distribution from a large set of lower-level data points and avoid difficulties that may arise from estimating such a distribution from intermediate levels that comprise only a small number of data points (3 experiments or 4 monkeys). However, it is very interesting to investigate which models wins the model comparison at the level of individual experiments as well as on the level of individual monkeys. For this reason, we now also repeat our model comparison for each monkey separately by considering all sessions from the same monkey irrespective of experiment. As mentioned above, the results of this approach were highly consistent with our new behavioral ridge regression analysis. They show that a model that parameterises the global reward state better captures

behavioral choice patterns than a model without such a parameter. This is true for three out of four monkeys; the same ones in which we found behavioral evidence of the effect of the global reward state.

We clarified the model fitting procedure in the Methods in the following way:

Model fitting. All RL modelling was conducted in Matlab (version 2018a). We used an iterative expectation-maximization (EM) algorithm to fit the models^{27,28}. We estimated models both over all experiments and for each experiment **and monkey** separately. This ensured that our results were valid over all data sets but also within each experiment **and monkey**. It also enabled us to use the experiment-appropriate parameter estimates for the analysis of the MRI data. **In all cases the fitting comprised two levels: the lower level of the individual sessions and the higher-level reflecting either all of the sessions together, all sessions from the same experiment, or all sessions from the same monkey.**

In the main manuscript, the figure panels comparing best-fitting to second best model now also indicate monkey identity:

Figure 3.... (g,h,i) Experiment specific comparisons of Bayes factors between full model and second-best model. Bars are individual sessions and direction of the bars (left vs right) indicate better model fit in favor of the full (going to the right) compared to the second-best model (going to the left).

Colors indicate individual monkeys. (n = 65 sessions). Source data are provided as Source Data file.

Finally, we include a comprehensive report of the model comparisons on the individual monkey level in the supplements. Note also that we provide a detailed source data file together with this manuscript that supplies tables with all numeric values displayed in the following panels:

Supplementary Figure 3. Supplementary model comparison over experiments and monkeys. For these analyses, the sessions were resorted according to the experiment they were recorded in irrespective of monkey identity (a-f) or according to the identity of monkey irrespective of experiment (g-r). (a-f)

Supplementary model comparison for individually fitted experiments. BICint and exceedance probability (XP) for the fMRI data set (a,d) and the two behavioral data sets (b,e and c,f, respectively). According to both indices, the full model best explained the observed choice patterns among the set of tested models. The XPs of the full model are: XP-FMRI = 0.990; XP-BEHAV = 0.997; XP-BEHAV2 = 0.999. Note that the log Bayes factors for the best-fitting compared to the second-best model for each experiment are displayed in the main manuscript. (g-r) Supplementary model comparison for individually fitted monkeys. BICint, XPs, and log Bayes factor of the overall best-fitting compared to the second-best fitting model are shown in separate columns for monkey 1 (g-i), monkey 2 (j-l), monkey 3 (m-o) and monkey 4 (p-r). The x-labels are the same for all panels of the same column and are displayed in the last row. For monkeys 1,2, and 4, both the BICint and the XP clearly favour the full model. The XPs of the full model are: XP-monkey1 = 0.973, XP-monkey2 = 0.998, XP-monkey4 = 0.999. By contrast, for monkey 3, there are divergent results when considering the XP or the BICint. Regarding the XP, no model reaches an XP higher than 0.95; the relative best model for monkey 3 is one that does not contain a reward trace (XP-monkey3-RL+cl+cs=0.625), followed by two models that contain a reward trace (RL+cs+rt and RL+cl+cs+rt). By contrast, according to the BICint, the marginally best model is one that contains a reward trace (BICint-monkey3-RL+cs+rt= 2543.8) followed by one that does not contain one (BICint-monkey3-RL+cl+cs=2549.4). Note that monkey 3 is the same animal for which there was no clear effect of the global reward state in the corresponding behavioral analyses (R-history effect in Supplementary Fig.2k), while the remaining three animals clearly showed such an effect. In conclusion, there is no clear winning model when considering both the BICint and the XP together for monkey 3. However, it suggests that monkey 3's choices were less influenced by the global reward state compared to the other monkeys. However, all together and in accordance with the behavioral analyses of individual monkeys, these results clearly demonstrate that the global reward state effects described in this manuscript are not driven by single outlying individuals. Note that the precise BICint and XP values for all panels are provided in the source data file related to this manuscript. The dashed red line in panels displaying exceedance probabilities indicates an exceedance probability of 0.95. Source data are provided as Source Data file.

- The authors do a thorough job in comparing models with all the possible permutations of CL, CS and R-trace. It wasn't clear to me, however, that the effect of GRS could only be exerted as influence on the PE. For example, perhaps motivation to switch is lower when GRS is higher (lower need for exploration)? In other words, is it possible that GRS affects the weighting of CL-trace and/or CS-trace? Also, the authors stress that their model can reproduce dynamic changes in value update rates and asymmetries in value updates without explicitly adjusting learning rates. Is it possible to explicitly compare with a model that does adjust learning rates? Can you explain why one is more likely than the other?

This is a complex question. It does go beyond the rationale of our current modelling approach, which we derive from our behavioral analyses in which we found evidence for non-contingent choice and reward traces in the first place. We answer this question by addressing the three sub-questions contained in it. The first one is whether a category of alternative models fits the data better in which the influence of the R-trace/GRS is not mediated by the PE but instead by the choice traces. The second one is whether another category of alternative models might fit the data better that does not assume a R-trace but instead uses more common asymmetric or dynamic learning rates. And the third question (which is very related to the second one) is why the value update dynamics in the data are more specific to our model than to others. To answer these questions, we fitted over 10 new control models, none of which fitted the data better than our model. We added descriptions of these 10 models to the Methods section. We present the results of the new model comparison in the supplements as well as parts of it in the main text and in a modified main figure. We estimate the *effective learning rates* from our full model to demonstrate that the value updates change specifically depending on the GRS and this is the conceptual explanation why our model is more likely than others. We have added all of these control models to the online repository together with the other Matlab RL code so that it is possible for everyone to track the implementation of the models. Below we answer the three

sub-question in detail and afterwards show the changes in main text, methods, supplements, main figures and supplementary figures that this has led to.

1) To start with, in our current model comparison, we consider models with a choice location trace (CL-trace), a choice stimulus trace (CS-trace) as well as a reward trace (R-trace) that modulates PEs. We consider all possible permutations of these mechanisms and find that a model including all three memory traces (RL-cl+cs+rt) fits the observed behavior best. We refer to this model also as the “full model”. The first category of control models assumes, just as the full model, an effect of R-trace, but one that is conveyed not via a modulation of PEs but instead via affecting non-value-related mechanisms such as the CL-trace or CS-trace weights. Such a modulation of choice memories could in principle generate behavior similar to the one we had observed – that animals increasingly stay with rewarded choices in a high GRS, but preferably switch away from options in low GRS environments. Such modes of action of R-trace are possible, albeit somewhat unlikely in our view. This is because, whereas the effects of the reward trace as part of the PE calculation remains within the realm of value in the full model, such an alternative account would imply that non-contingent reward memories can only be pressed into action in interaction with non-contingent choice memories. While this is not impossible, it would require us to always first assume that decision making is influenced by non-contingent choice memories and that additional effects of non-contingent reward traces are always secondary to and mediated by these choice memories. However, we have constructed such control models and show empirically that they did not fit as well as the full model. Following a suggestion by reviewer 3, we also consider that R-trace might affect the beta parameter of the softmax function and we constructed respective control models to address this idea. These control models belong to the same category, because they also assume an effect R-trace, but one that is not mediated by the PE.

We constructed 8 control models in this category. We considered that R-trace modulates the CL-trace, CS-trace, both CL- and CS-trace, or the softmax temperature beta. These are four possible alternative mechanisms and for each we constructed two versions of control models, one multiplicative and one additive. In the multiplicative version, we multiplied the respective weight with R-trace. This meant that e.g. the tendency to repeat the same stimulus choice is higher with a high R-trace and lower with a low R-trace. These models do in fact have the advantage compared to the full model that they use one less free parameter (w_R). However, note also that in these models, the impact of choice memory traces is entirely dominated by R-trace, meaning that if R-trace is zero, any tendency to repeat choices is abolished. This might not reflect the way in which R-trace modulates choice traces; instead, it might be more plausible to assume a constant effect of choice repetition that is then to some degree upscaled or downscaled by R-trace. This is why we constructed the additive control models. These make use of a free parameter that scales the additive impact of R-trace in addition to the other weight. We tested all of these models against the full model and, as mentioned before, none of them fits better than our full model.

However, we want to mention that we examined the free parameter values from these additional models, and we observed that they behaved as predicted by the reviewer. For instance, the model modulating the CS-trace exhibited a positive additional weight of R-trace meaning that the tendency to repeat choices was increased when R-trace was high. Since the additional control models did not fit the data as well as the full model and due to space limitations, we did not expand on these considerations in the manuscript, but we would be happy to do so if the reviewer feels this would be worthwhile. We also want to note that this category of control models is in principle compatible with our approach and the conclusions in our manuscript, because all of them do predict an influence of the GRS on choice in the same manner we observed in behaviour.

2) By contrast, the second category of control models does not require a reward trace at all. As highlighted by the reviewer, the full model effectively implements asymmetric (i.e. different for positive and negative outcomes) and dynamic (depending on whether R-trace is low or high) value updates. Therefore, we tested two new control models that implement either asymmetric or dynamic learning rates. The model with asymmetric learning rates uses separate learning rates for rewarded and for unrewarded trials. The model examining dynamic learning rates is based on the widely held idea that learning rates should increase when surprising outcomes are encountered, whereas it should decrease if outcomes conform to expectations. One RL implementation of this idea tracks the slope of absolute prediction errors to upregulate or downregulate the learning rate (Krugel et al. 2009, Queirazza et al., 2019). Note that this model is not, as the others, nested within more basic models. We implemented both models and compared them to the full model and found that the full model was a better explanation of the data both in terms of BICint and exceedance probability.

3) Finally, we looked into ways to improve on the conceptual explanation of why the dynamics of value updates are better explained by our model than by one that just assumes dynamic or asymmetric values updates. The key point about our model is that it makes specific predictions about *when* value updates should be low or high. It predicts that this depends on the nature of the outcome (positive or negative), but also on whether the current GRS is high or low. To demonstrate this, we derived *effective learning rates* from our model. This means we estimated the effect that the GRS in our model had on the speed of learning. We demonstrated this in a new modelling analysis in which we fit separate learning rates depending on whether outcomes are positive or negative and depending on whether the GRS is high or low. Then we test the specific prediction from our model that we should observe an interaction effect in these effective learning rates, and this is exactly what we did in our analysis. Note that this analysis is unusual in that the new model does not explicitly use R-trace, but capitalises on the knowledge of when the GRS is high versus low. This demonstrates that reference to the GRS is crucial to understand the dynamics of value updates in our data.

We changed the main text, methods, supplements, and respective figures in the following to convey these new analyses. Note that we added a model comparison regarding the new value models (point 2) and the effective learning rate analysis (point 3) to the main text, because we felt they were particularly useful in demonstrating and explaining the dynamics of our model. We added a model comparison including the non-value-R-trace models (point 1) to the supplements, because we felt this line of argument is a bit more remote to our main claims.

Main text and main figure:

.... This means that the model effectively implements asymmetric learning rates for positive and negative outcomes that change dynamically as a function of GRS; the impact of newly encountered outcomes is scaled up or down depending on the GRS. **We demonstrate this by estimating the *effective learning rates* from our model (see Methods) and indeed find a significant interaction depending on outcome type and GRS being low or high (Fig.3e; $F_{1,64} = 18.357$; $p < 0.001$).** This maps directly onto the pattern of stay-switch choices observed in the initial behavioral analysis, showing that a high GRS promotes win-stay and a low GRS promotes lose-switch behavior (Fig.1g). **It also explains why our model fits better than models assuming different learning rates for positive and negative outcomes (“AsyAlpha”) or models assuming dynamic learning rates based on the degree of surprise encountered over time (“dynAlpha”); Fig.3f,g; see Methods for details of the modelling). The reason is that reference to the GRS is critical to explain the dynamics of value updates in our data.**

Figure 3. R-trace related learning dynamics. (a) Posterior density over w_R for the FMRI experiment as well as the two behavioral experiments. (b) Percentage of posterior distribution shown in panel a that is bigger than zero. (c) Average PEs binned by current outcome (Reward/No reward) and R-trace (low/high; median split) show the offset of PE coding generated by positive w_R . (d) Absolute PEs plotted as in panel c illustrate stronger positive value updates when R-trace is high and stronger negative value updates when R-trace is low. (e) Estimated *effective learning rates* of the model demonstrate that the speed of learning differs based on both the type of current outcome and the GRS. (f,g) BICint and exceedance probability indicate that the full model fits better than one using either asymmetric or dynamic learning rates. The dashed red line indicates an exceedance probability of 0.95.

Supplements:

Supplementary Figure 4. (a-i) ... (j,k) (j,k) Supplementary control model analyses showing BICint and exceedance probabilities for the full model as well as all supplementary models. The model comparison includes all supplementary models as explained in the Methods section. There are two broad categories of control models: models that assume an influence of R-trace, but one that is not mediated by PEs but by different mechanisms; and a second category that assumes no R-trace, but instead that value updates have different dynamics. The full model RL+cl+cs+rt is the winning model according to both BICint and exceedance probability (exceedance probability = 0.959). The red line indicates an exceedance probability of 0.95. Source data are provided as Source Data file

Methods:

Supplementary control models. In our main model comparison, which follows on from our behavioral analyses, we formalize the effects of non-contingent choice and reward memories on decision making in an RL framework. We consider models with a choice location trace (CL-trace), a choice stimulus trace (CS-trace), and a reward trace (R-trace) that modulates PEs. We consider all possible permutations of these mechanisms and find that a model including all three memory traces (RL+cl+cs+rt) fits the observed behavior best. We refer to this model also as the “full model”. However, in a supplementary model comparison we also consider two additional categories of control models.

The first category assumes, just as the full model, an effect of R-trace, but one that is conveyed not via a modulation of PEs but instead via affecting non-value-related mechanisms such as the CL-trace or CS-trace, or the β parameter of the softmax function (equation 4). Such a modulation of choice memories could in principle generate behavior similar to the one we had observed – that animals increasingly stay with rewarded choices in a high GRS, but preferably switch away from options in low GRS environments. Similar effects are possible if R-trace modulated the β parameter because it might increase or decrease random exploration at times of low or high GRS. Such modes of action of R-trace are possible, albeit somewhat unlikely in relation to the choice traces. This is because, when reward trace exerts its impact via the PE calculation its effects remain within the realm of value in the full model. However, the alternative account would imply that non-contingent reward memories can only be pressed into action in interaction with non-contingent choice memories. While this is not impossible, it would require us to always first assume that decision making is influenced by non-contingent choice memories and that additional effects of non-contingent reward traces are always secondary to and mediated by these choice memories. Nevertheless, we tested a comprehensive set of alternative models that incorporated such R-trace mechanisms. These alternative models mimicked the full model but eliminated the effect of R-trace from the PE calculation (equation 14) and thereby also removed the free parameter w_R that determined the weight of R-trace on the PE. Instead, in the alternative models, R-trace modulated the weights of CL-trace, CS-trace or both. We denote the new models by the interaction type of R-trace with the respective choice trace, e.g. “RT \times CL”, “RT \times CS” or “RT \times CL&CS”. We implemented the new models by changing equation 13 in the following ways (note that we explicitly write down the multiplication symbol as \times for clarity):

$$23) DV_{RT \times CL,t} = DV_{RLsimple,t} + w_{CL} \times R\text{-trace}_t \times CL\text{-bonus}_t + w_{CS} CS\text{-bonus}_t$$

$$24) DV_{RT \times CS,t} = DV_{RLsimple,t} + w_{CL} CL\text{-bonus}_t + w_{CS} \times R\text{-trace}_t \times CS\text{-bonus}_t$$

$$25) DV_{RT \times CL\&CS,t} = DV_{RLsimple,t} + w_{CL} \times R\text{-trace}_t \times CL\text{-bonus}_t + w_{CS} \times R\text{-trace}_t \times CS\text{-bonus}_t$$

These models mediate the impact of R-trace on the decision variable via the choice memory traces. These models do in fact have the advantage compared to the full model that they use one less free parameter (w_R). However, note also that in these models, the impact of choice memory traces is entirely dominated by R-trace, meaning that if R-trace is zero, any tendency to repeat choices is abolished. This might not reflect the way in which R-trace modulates choice traces; instead, it might be more plausible to assume a constant effect of choice repetition which is then to some degree upscaled or downscaled by R-trace. For this reason, we constructed additional similar control models that assume an additive effect of R-trace on the choice memory trace weight via an additional free parameter. We denote these models with “RT \times CL 2”, “RT \times CS 2” or “RT \times CL&CS 2”:

$$26) DV_{RT \times CL 2,t} = DV_{RLsimple,t} + (w_{CL} + w_{RT \times CL} \times R\text{-trace}_t) \times CL\text{-bonus}_t + w_{CS} CS\text{-bonus}_t$$

$$27) DV_{RT \times CS 2,t} = DV_{RLsimple,t} + w_{CL}CL\text{-bonus}_t + (w_{CS} + w_{RT \times CS} \times R\text{-trace}_t) \times CS\text{-bonus}_t$$

$$28) DV_{RT \times CL \& CS 2,t} = DV_{RLsimple,t} + (w_{CL} + w_{RT \times CL} \times R\text{-trace}_t) \times CL\text{-bonus}_t + (w_{CS} + w_{RT \times CS} \times R\text{-trace}_t) \times CS\text{-bonus}_t$$

In close analogy to these control models assessing a mediation of R-trace effects via choice memory traces in a multiplicative or additive fashion, we also considered whether R-trace acts by modulating the inverse temperature parameter from the softmax function (equation 4). We constructed two new models, again without and influence of R-trace on the PE calculation. Instead, R-trace modulated the β parameter of these models. The new β parameters for these models were constructed as:

$$29) \beta_{RT \times \beta,t} = \beta \times R\text{-trace}_t$$

$$30) \beta_{RT \times \beta 2,t} = \beta + w_{RT \times \beta} \times R\text{-trace}_t$$

All of the above models belong to the first category of supplementary control models we examined. They all have in common that they assume an effect of R-trace on choice. However, they all assume that this effect is mediated via different routes compared to the one we use in the full model, i.e. instead of influencing value update in the PE, they assume that R-trace acts on non-value-related mechanisms of the model.

By contrast, the second category of control models that we examined does not require a reward trace at all. As explained in the main text, the full model effectively implements asymmetric (i.e. different for positive and negative outcomes) and dynamic (depending on whether R-trace is low or high) value updates. The first control model from this category therefore examines whether a model with no reward trace, but instead asymmetric learning rates^{48,70} fits the data better than the full model. This model (“AsyAlpha”) makes use of the architecture of the full model without R-trace. This also removed the weight parameter, w_R , of R-trace, as well as the R-trace specific learning rate α_R . Instead, the AsyAlpha model uses separate learning rates for rewarded and for unrewarded trials. The final control model is also similarly based on the architecture of the full model without R-trace but assumes a dynamic learning rate (“DynAlpha”). It is based on the widely held idea that learning rates should increase when surprising outcomes are encountered, whereas it should decrease if outcomes conform to expectations. One RL implementation of this idea tracks the slope of absolute prediction errors to upregulate or downregulate the learning rate^{29,71}. Note that this model is not, as the others, nested within more basic models. It uses the free parameter α_R as the initial learning rate on the first trial, but also to control the smoothing when tracking the absolute prediction errors (PEmag):

$$31) PEmag_t = (1 - \alpha)PEmag_{t-1} + \alpha|PE_t|$$

From this, the normalized slope of absolute prediction errors (m) is calculated:

$$32) m_t = \frac{PEmag_t - PEmag_{t-1}}{(PEmag_t + PEmag_{t-1})/2}$$

The slope is then used to generate a link function using a free parameter γ :

$$33) fm_t = sign(m_t) \left(1 - e^{-\left(\frac{m_t}{\gamma}\right)^2}\right)$$

And this link function is then used to calculate the learning rate (LR) for the current trial:

$$34) LR_t = \begin{cases} LR_{t-1} + fm_t \times (1 - LR_{t-1}) & \text{if } m_t \geq 0 \\ LR_{t-1} + fm_t \times LR_{t-1} & \text{if } m_t < 0 \end{cases}$$

Effective learning rate calculation. As noted in the main text, the effect of R-trace in the full model RL+cl+cs+rt is that it impacts the PE calculation. This means that it increases or decreases the PE based on the current GRS, which will increase or decrease the change in Q-value. Such a mechanism can produce asymmetric and dynamic changes in value updates without explicitly changing the learning rate. Our model predicts that value updates from the same outcome event are higher for rewarded trials when R-trace is high compared to low and vice versa for unrewarded trials. We specifically tested this hypothesis by calculating the *effective learning rates* from our full model. We did this by running an additional model analysis. For this analysis, we fitted a new model that did not contain an explicit R-trace. In addition, we fixed the free parameters for each session to the session-specific parameter values from the full model. The model comprised four free parameters, all of which were learning rates, but different ones for rewarded and unrewarded trials, and ones with a high (i.e. ≥ 0.5) or low (i.e. < 0.5) R-trace. Note that the R-trace information was session-specific and imported from the full model; it was not calculated or fitted in effAlpha itself. In other words, what this model did was to assume that different effective learning rates were used for positive and negative outcomes depending on high or low R-trace. It keeps all other features identical to the full model by fixing the remaining parameters to the ones from the full model. But instead of using R-trace in the PE calculation, it examines whether the effective learning rates indeed differ in the manner expected by our full model. Note that in order to test the learning rates from effAlpha for significant differences, we fitted the model session-wise via maximum likelihood estimation, because the hierarchical fit decreases the variance of individual parameters which possibly biases comparisons of free parameters.

- Similar to the behavioral analysis, in the imaging analysis it was also not clear to me how the four individuals were accounted for. The methods section mentions that session was treated as a random effect – but there were multiple sessions per monkey, doesn't that inflate the significance of the results? Can you show results within monkey?

We indeed ran our fMRI analyses in two levels. A first one with session-specific GLMs and a second-level analyses in which we ran a random effects analyses over all sessions irrespective of monkey identity. This analysis approach is identical with previously published fMRI investigations in macaques and to our knowledge does not inflate the significance of the results. Given the structure of our data, it is unfortunately not possible to have an intermediate subject level on the second level and then to do a random effects analysis over subjects on the third level. This is because a random variable is required to have at least 10 levels to estimate random effects (we have 4 monkeys) (Raudenbush and Bryk, 2001). But for both practical and ethical reasons, it was impossible to test more than four macaques and in fact our subject group already exceeds the size of most non-human primate studies. We concede this limitation in our methods section:

Using this framework we initially performed a first-level fixed effects analysis to process each individual experimental run. These were then combined in a second-level mixed-effects analysis (FLAME 1) treating session as a random effect (we also had a similar number of sessions across subjects) and using family-wise error cluster correction ($z > 2.6$ and $p = 0.05$). **One reason for omitting an intermediate level of MRI analysis was that the subject number in our study, as in the majority of studies of macaque neural activity, was below the one required for random effects analyses across individual monkeys⁷⁸.**

In addition we did consider whether our key results were driven by data from an individual monkey. We tested whether the neural effect sizes differed across monkeys using a one-way ANOVA with a factor of monkey identity. We did not find any evidence for a difference in effect sizes over monkeys for our key analyses. In addition, we changed the display of these figures to make the identity of the monkeys the data is from even clearer. We use the same symbols that we have used in our behavioural and modelling analyses. We added this to the manuscript as follows :

We found that, strikingly, Ia represented *R-trace* during both rewarded and non-rewarded trials in the left (rewarded: $t_{24} = 2.07$; $p = 0.049$; unrewarded: $t_{24} = 3.68$; $p = 0.01$) as well as in the right hemisphere (rewarded: $t_{24} = 3.44$; $p = 0.002$; unrewarded: $t_{24} = 3.64$; $p = 0.001$; Fig. 4c,d). There was no significant difference between conditions in left ($t_{24} = 1.23$; $p = 0.23$) or right Ia ($t_{24} = 0.73$; $p = 0.47$), nor did the effects differ between monkeys in the left (one-way ANOVA on averaged effect sizes: $F(3,21) = 1.206$; $p = 0.332$) or right Ia ($F(3,21) = 2.195$; $p = 0.119$).

However, unlike Ia, DRN activity only reflected *R-trace* on non-reward outcome trials, but not on rewarded trials ($t_{24} = 0.51$; $p = 0.62$). There was a significant difference between conditions ($t_{24} = 2.96$; $p = 0.007$; Fig.6b; using a leave-one-out DRN ROI to avoid bias). These effects of *R-trace* during unrewarded trials did not differ across monkeys : $F(3,21) = 2.403$; $p = 0.096$).

- The authors report activation patterns that are consistent with their model, but also with other models. They do not search for encoding of PE, which could be very helpful here. If activity in areas like the ventral striatum, where PE is usually observed, is indeed modulated by R-trace, it will support the particular model the authors put forth.

We appreciate the request for more neural specificity in the results. However, we tested several neural predictions from our model and we think that the neural results presented already align with our RL model in several important ways. First of all, the main prediction from our RL model is that the brain should hold a representation of the global reward state, i.e. *R-trace*. The fact that such a quantity is

indeed encoded in neural activity in several brain regions is a striking finding in itself. The brain regions that we have identified with the global reward trace, agranular insula, dACC and the dorsal raphe, also encode immediate reward outcomes. The model predicts that reward prediction errors are modulated by both immediate outcomes and R-trace -- the two signals that we show are present in the areas. In particular the way in which these two quantities are represented in the insula maps directly on the way they are combined in the model.

In theory, we should be able to take this one step further. The reward prediction error, according to the model, depends not just on the immediate outcome and R-trace but also on a third variable – the chosen value – the expected value of the choice taken. There is a problem with this variable. We know from both electrophysiological recording and fMRI studies that it modulates activity in prediction error encoding brain areas in two opposite directions at two different time points. At the time that the choice is made, reward expectation leads to increased neural activity; higher reward expectations are associated with greater brain activity. At the time of reward receipt, however, higher reward expectation leads to decreased neural activity (this might need a moment's reflection to consider but this is why an anticipated reward that is not delivered is associated with decreased activity and why an anticipated reward leads to no activity – the positive impact of the reward outcome is balanced by the negative effect of the prior reward expectation). In the current experimental design, however, we cannot examine the sign change in activity associated with reward expectation because the interval between choice and outcome in the present design is not sufficiently long. As chosen value is often represented with different signs at the time of decision compared to the time of outcome, we chose to disregard chosen value for the purpose of the neural analysis, as its interpretation would not have been straightforward and on the contrary might even have biased our results. We have tried to convey this line of reasoning more clearly at several places in the manuscript:

One of the main predictions from our winning RL model (RL+cl+cs+rt) is that the brain should hold a representation of the GRS. Therefore, we started our MRI analysis by asking whether there is a brain network that codes the GRS, i.e. the R-trace variable from our winning model. In a first GLM (GLM1) we indeed found such signals. We found the strongest evidence of encoding of R-trace in bilateral anterior insular cortex at the time of choice feedback (cluster-corrected at $Z > |2.6|$, $p = 0.05$ as all whole-brain results reported in this manuscript), just posterior to medial orbitofrontal cortex, which we refer to as agranular insula (Ia; Fig.4a; Supplementary Table 1)...

A key feature of our RL model is that reward prediction errors driving learning are informed by both immediate outcomes and R-trace. These two sources are integrated with the same sign in the value update: an option's value increases if it is rewarded but also if GRS is high. Therefore, it is notable that current outcome and R-trace both affect activity in Ia and that they do so with the same sign. Another key feature of the model is that R-trace has a positive effect on choice value updates during both rewarded and unrewarded trials.

We found that, strikingly, Ia represented R-trace during both rewarded and non-rewarded trials in the left (rewarded: $t_{24} = 2.07$; $p = 0.049$; unrewarded: $t_{24} = 3.68$; $p = 0.01$) as well as in the right hemisphere (rewarded: $t_{24} = 3.44$; $p = 0.002$; unrewarded: $t_{24} = 3.64$; $p = 0.001$; Fig. 4c,d). There was no significant difference between conditions in left ($t_{24} = 1.23$; $p = 0.23$) or right Ia ($t_{24} = 0.73$; $p = 0.47$), nor did the effects differ between monkeys in the left (one-way ANOVA on averaged effect sizes: $F(3,21) = 1.206$; $p = 0.332$) or right Ia ($F(3,21) = 2.195$; $p = 0.119$). Note that we refrain from analyses of chosen value in this line of investigation. Although chosen value also informs the reward prediction errors in addition to outcome and R-trace, the interpretation of this variable is less unequivocal due the fact that there is only a short time between decision and outcome in our experiment.

- The authors mention that in other contexts they and others have observed contrasting, rather than facilitating, effects of GRS. Could you discuss the different contexts, when is GRS weighted positively and when negatively? How do the results relate (or not) to previous work about menu normalization effects?

It is indeed a very intriguing question when the global reward state will have a positive effect, a negative effect, or no effect at all. Whether and how this relates to menu invariance in choice values is not clear. We are wondering about this, too, and this is in our view a central question for subsequent investigations into such effects. Species differences might play a role in this. Many of the cases of positive GRS effects (such as in this study) were found in macaques, whereas negative GRS effects have been found in humans. However, task differences may be even more important. One possibility is that positive GRS effects are found particularly in cases when subjects have difficulty separating one trial from subsequent ones and hence increase or decrease the value of their choices based on a longer-term sense of the success of their actions. It might be that humans are much better in separating an experiment in discrete trials compared to macaque monkeys, particularly if the macaques such as ours are not yet overtrained. However, it is possible to design experiments in which trial separation becomes unclear even for human subjects (eg Jocham et al., 2016) and in such situations we anticipate that positive GRS will also occur. We now articulate these lines of thought more clearly in the discussion as follows:

A feature of the model is that the direction of influence of the GRS on learning is determined empirically by fitting w_R . That influence can be positive or negative or absent. The positive effects of GRS observed in our study are in line with previous reports of spread of effect^{8,11-13}. They suggest an RL mechanism mediating the “misassignment” of previous rewards to subsequent choices. Negative w_R , on the other hand, implements a temporal contrast effect whereby the value of new outcomes is de-weighted in a high GRS, because they are referenced to the already high value of the environment. Such contrast effects have been reported in other contexts^{17,18,31,47}. It is an open question which task features incentivize positive or negative effects of GRS. Furthermore, species differences might also play a role as many of the cases of positive GRS effects (such as in this study) were found in macaques, whereas negative GRS effects have been found in humans. One possibility is that positive GRS effects are found particularly in cases when subjects, even human participants, have difficulty separating one trial from subsequent ones⁸ and hence increase or decrease the value of their choices based on a longer-term sense of the success of their actions.

Reviewer #2 (Remarks to the Author):

This paper proposed a new RL model in which the calculation of reward prediction error is biased by a number tracking the recent average reward rate (global reward state). This bias causes the RL agent to have a higher chance of repeating the actions being rewarded in a time period when the agent has often been rewarded regardless of the choices (rich context), and a higher chance to avoid unsuccessful choice in a poor context. In addition to explaining monkey behavioral choice, the paper also identified anterior insular and dorsal raphe nucleus as regions that track this global reward state. In contrast, neural signal in vmPFC only reflects the value difference between the two choices, but not the global reward state.

To my knowledge, the idea of a global reward state is quite novel, and the implication is intriguing: instead of assigning credit of current reward to some choices made in the past, monkeys appear to in fact assign credit of past reward to current choice. This is quite a surprising and interesting finding.

The experiment design is beautiful. The analysis procedure appears generally solid, except for a few confusions in the method (see below). The paper is well written. Graph illustrations are clear. Major codes and behavioral data are shared. I suggest the authors considering sharing the fMRI data as well.

We are happy about the positive evaluation of the reviewer and we really appreciate that they had read our paper with such care. We have added the group-level contrast images that we show in our manuscript to an online repository and provided the link to it together with our revision. We have updated the Data availability statement accordingly:

Data availability. We have deposited all choice raw data in an OSF repository. All reinforcement learning results in this paper are derived from these data alone. The accession code is:

https://osf.io/358cg/?view_only=0e6fda7925364d86930374cd4ae4a59f

Any remaining data that support the findings of this study are available from the corresponding author upon reasonable request. **We have also deposited all group-level contrast images presented in the manuscript on Neurovault. The accession code is:**

<https://neurovault.org/collections/KJVDIJYY/>.

Sharing the entire data set is difficult but we think that the recently formed PRIME-DE consortium might provide a way to do this. We have released many other data sets to this consortium in the past and we will investigate whether it is possible to release these data too (unlike previous data held by PRIME-DE, the current data set is not a resting state data set but rather a behavioural task-related data set). We thank the reviewer for spotting a number of inconsistencies in our manuscript and suggesting improvements for conveying the modelling results. We addressed these points in our revision.

I generally recommend acceptance, but I think minor revision to address the points below would enhance the paper:

1. Online Method, Page 27, EM procedure. Based on the description, it feels to me the procedure is actually coordinate descent instead of EM. I don't see a clear marginalization procedure typical in the expectation step of EM. Equations 15 and 16 are inconsistent with the paragraph above. In the equations, there is a sum of prior probability over μ and σ , but the paragraph seems to suggest only a single value of μ and σ are used. Indeed, Equations 17 and 18 specify how point estimates of μ and σ are calculated. And to me,

it also makes no sense to sum the log prior probability in the second term of equation 15 and 16.

The acronym NPL seems to have no relation with the term "maximum posterior likelihood". Also, I suggest using "posterior probability" instead of "posterior likelihood" (think of prior, likelihood and posterior in Bayesian framework: likelihood has a specific meaning and it is better not to use it to replace "probability" in a general sense).

To be more accurate, I also suggest using $p(\text{choice}_t | h, \text{choice}_{\{1, \dots, t-1\}})$ in place of $p_t(\text{choice} | h)$. Strictly speaking, the likelihood of each trial is not independent, only the conditional probability of the current trial given the DV (which depends on all the past choices, past stimuli and h) is independent across trials and thus can be multiplied.

It may be better using Latin letters μ and σ instead of the words *mu* and *sigma*.

We thank the reviewer for spotting some inconsistencies in the terminology as well as in some equations. We have corrected these mistakes, and provide a point-by-point response below.

First, we followed the reviewer's suggestion to write μ and σ^2 instead of using the words *mu* and *sigma*.

Second, we agree with the reviewer that the EM algorithm typically involves a marginalization procedure over different "classes" (i.e., different Gaussian distributions in the case of a mixture of Gaussian distributions). However, here and as in Huys et al., 2011, 2012, we do not use the EM procedure to classify the parameters of each session as belonging to different classes from a mixture of Gaussian distributions. Instead, there is a single Gaussian distribution representing the group-level prior over parameters, and we search for μ and σ^2 of this distribution which maximizes the posterior. This is why we do not need a marginalization procedure over classes.

We think that this is nevertheless different from coordinate descent because the descent for each dimension of the parameter vector is simultaneous, rather than being sequential. Here we assume independence between model parameters, like the mean field assumption in variational Bayes.

Third, we followed the reviewer's suggestion to rename NPL to PP for Posterior Probability.

Four, the reviewer is right about $p(\text{choice}_t | h, \text{choice}_{\{1, \dots, t-1\}})$. In the code, we do add the logs of conditional probabilities of current trial given DV and model parameters h , which are independent across trials given the Markov property assumed by the model. Now this gives an approximation to the likelihood function $p(\text{all choices} | \text{model}, h)$, which is why we thought we could simply abbreviate things by writing $p(\text{choice}_t | h)$. But following the reviewer's remark, we now abbreviate it with $p(\text{choice}_t | \text{DV}, h)$ (DV referring to decision variable) to be more accurate.

Finally, the reviewer is right that, in the paragraph above equations 15 and 16, we described that μ and σ^2 as point estimates (one μ and σ^2 per session) whereas in the equation, we wrote a sum over μ and σ^2 as if we were integrating over these values. This was a mistake and we thank the reviewer for catching it. Indeed, the previous equation did not reflect what was actually implemented in our code. The reason why we wrote a sum in the second term of equations 15 and 16 is because h is a vector of several model parameters (several independent dimensions, as discussed above). So, in the code we calculated separate normpdfs per parameter and then summed their logs to compute the prior (but of course given a single μ, σ^2 per parameter, which represents the group-level prior distribution over that parameter).

The second term of equations 15 and 16 simply represents the prior probability of the vector of parameters h given group-level Gaussian distributions over the parameters with a mean vector μ and standard deviation σ^2 . We thus corrected the equations in the paper accordingly:

$$15) PP_i = \max_h \left[\sum_t (\log(p_t(\text{choice}_t | DV, h_i)) + \log(\text{normpdf}(h_i | \mu, \sigma))) \right]$$

$$16) h_i = \text{argmax}_h \left[\sum_t (\log(p_t(\text{choice}_t | DV, h_i)) + \log(\text{normpdf}(h_i | \mu, \sigma))) \right]$$

In order to be more precise in the description of the model fitting, we rewrote the paragraph above equations 15 and 16 in the following way:

During the expectation step, we calculated the log-likelihood of the subject's series of choices given a model M and its parameter vector h_i for session i (i in $\{1..N\}$). To do so, we summed the conditional probability of each trial's choice given the model's DV and parameters h_i (here abbreviated $\log(p(\text{choice}_t | DV, h_i))$) over all trials of a session. We then computed the maximum posterior probability (PP_i) estimate obtained with this parameter vector h_i given the observed choices and given the prior computed from group-level Gaussian distributions over the parameters with a mean vector μ and standard deviation σ^2 .

2. Is the Hessian matrix calculated based on just the likelihood (which does not include prior), or the posterior? If posterior, is it based on the initial prior (mu and sigma), or the recalculated mu and sigma? I feel that the Hessian of the likelihood or the posterior using the initial prior may be the correct way, at least for the purpose of Equation 22. Please state it clearly.

We indeed calculated the Hessian matrix based on the posterior using the likelihood and the initial prior for the given iteration. After that, we recalculated mu and sigma and used these as priors for the next iteration step. We hope that we have clarified some of this in our previous response, but also added some clarifying remarks to equation 22. In addition, following point 11 of the reviewer below, we have also added information about mu and sigma being point-estimates to the main text :

Methods

As a second index of model fit, we used the Laplace approximation to calculate the log model evidence (LME) per session i based on the posterior probability PP_i (see equation 15):

$$22) LME_i = -PP_i - \frac{1}{2} \log(\det(H_i)) + \frac{Np}{2} \log(2\pi)$$

Note that the hessian matrix was calculated based on the posterior estimates using the likelihood and the initial group-level prior estimates. We submitted the LME scores to `spm_BMS`³² to compute the 'exceedance probability', the posterior probability that one model is the most likely model used by the population among a given set of models.

Main text :

However, the direction of the influence of GRS critically depends on w_R . In principle, the model allows GRS to have a negative, contrasting effect ($w_R < 0$), no effect at all ($w_R = 0$), or a positive effect ($w_R > 0$). We therefore examined the fitted parameter values of w_R , i.e the final group-level point-estimates of μ and σ associated with w_R after fitting the full model separately for each experiment.

We found that the posterior density of the fitted w_R was overwhelmingly positive for all three experiments (Fig.3a).

3. Page 36, in neural model comparison, it is worth specifying how the log-likelihood used for Bayesian model selection is calculated. Is it marginal likelihood, or maximal likelihood?

We used the maximum log likelihood estimates provided by Matlab's *fitglm* function. We now specify this in the Methods as follows:

We used Matlab's *fitglm* function together with the convolved design matrix and the BOLD timecourse for this regression and extracted the maximum log likelihood estimates from the resulting model.

4. Figure 1g. I did not find a description of how the residual probability is calculated.

We used Matlab's *glmfit* function and then examined the residuals as provided by the output stats object. We realise that this analysis as well as the entire behavioural GLM analysis is relatively complex. For this reason, we have added the Matlab code calculating the behavioural regression as well as the residuals analysis to the online repository together with the RL code. In addition, we provide a source data file that compiles all raw values that are displayed in these figures. This GLM code produces a table that matches the values in this source data file. We changed the manuscript accordingly:

We then examined the residual choice probabilities (as provided by the stats object of Matlab's *glmfit* function),

5. Fig 5e, f: the ANOVA over so many time points without mentioning a control for multiple comparisons is worrying. Also, due to the upsampling, adjacent time points would be highly correlated and redundant. So the F-statistics and p-value should really be just for illustrative purposes and not for drawing conclusions. The lack of control for multiple comparisons should be clearly stated.

We are aware of this feature of the ANOVA over time points. We refer to these panels in one instance to say that we don't observe an interaction effect anywhere in the timeseries. In regards to this, the features highlighted by the reviewer actually work *against* our conclusion rather than for us. And secondly we test the timecourse only for significance at one specific point in time – and that is when the effect in the contralateral hemisphere peaks. However, because we fully agree that this analysis is influenced by the autocorrelation in the BOLD timeseries and does not control for multiple comparisons, we now state this clearly and prominently in the main text:

Highly consistent with our model predictions, we observed no interaction of *R-trace* and outcome at any point in the time course (Fig.5e). We repeated the analysis for the right Ia and observed the same temporal pattern (Fig.5f). Even though, on the whole brain level, the outcome activity did not overlap directly with right Ia, we found a significant effect of outcome when examining the right Ia time course at the time of the contralateral peak of this effect (time = 6.15 seconds; $F(1,24) = 7.2$; $p = 0.013$). Note, however, that the series of p-values show in panels 5e,f are mainly for illustration as this analysis does not control for autocorrelation in the BOLD timeseries and multiple comparisons.

6. The caption for supplementary figure 1a says the result does not change with history length. I think it is worth specifying what range of history length has been tested.

We agree with the reviewer and made an additional supplementary figure that clearly demonstrates how the effect of interest, R-history, is stable when varying the time window between 2 and 9 trials. We added panel c to Supplementary Figure 1 to do this:

Supplementary Figure 1. Behavioral choice GLM.... (c) The effect of R-history on choice is stable across a broad window of reward history length. In our main GLM, R-history is calculated as the arithmetic mean of rewards occurring during the last six trials (see explanation in the legend of panel a). We repeated our main GLM and varied the length of this reward history between two and nine trials. The figure displays the significant effects of R-history in all of those GLMs. The blue circle indicates the R-history effect from the main GLM. The relatively weakest – but still significant – R-history effect was apparent when aggregating over nine trials ($t_{11} = 3.840$; $p = 0.003$). This might reflect the timescale over which the animals aggregate the general rewards state. However, it might also partly reflect the difficulties of estimating regression weights with a very large regressor set (the corresponding GLM contains 34 regressors). Note that varying the timescale of R-history makes it necessary to simultaneously adjust the timescale of the other relevant effects in the GLM. This will ensure that past choice-reward contingencies as well as choice history alone have a similar chance to account for the observed choices as R-history. For this reason, we concurrently varied the history length over which CxR-history and C-history were calculated. For example, when reducing the history length of R-history from six to five trials, we also reduced the history length of CxR-history and C-history by one trial (for both the current choice and the upcoming alternative choice; see description in panel a) and so on. Panels b,c concatenate

sessions per monkey per experiment resulting in three data points per individual; error bars are SEM across monkey data sets; $n = 12$; Source data are provided as Source Data file. Symbols indicate monkey identity in panels b,c; "MK" abbreviates "monkey".

7. Figure 2 had dashed line without explanation (it only appeared in the caption for Fig 7)

Many thanks for spotting this. The red line indicates an exceedance probability of 0.95 and we have added this information to the figure legend.

8. On Page 13 the term "residual BOLD" suddenly appears without any explanation of what it is. Although the procedure is explained in methods, it is worth briefly explaining what it is to help the reader understand.

Thanks, upon rereading we realised that the residual BOLD analysis is introduced very abruptly. We have rewritten the section to introduce the residual BOLD analysis more gently:

...

la, particularly left la (because of the extensive region in which strong *R-trace* and outcome effects overlapped), seemed to be the most likely brain region to integrate past and current rewards in a manner predicted by the computational model. In order to best understand neural activity in relation to behavioral and RL analyses, we regressed all effects of no interest out of the BOLD signal (i.e. all except *R-trace* and outcome effects) and then examined the residual BOLD signal (i.e. the variation in BOLD not explained by these regressors which we extracted from Matlab's stats.resid object). We found a striking temporal pattern of activity in left la when examining the residual BOLD time course in la as a function of both *R-trace* and outcome time-locked to the feedback onset (see Methods; Fig.5a,b,c,d,e). To this end, we binned it by *R-trace* (median split; low/high) and outcome (rewarded and unrewarded; *R* and *no R*) (compare with Fig.3c). Initially, residual BOLD activity clustered positively and negatively as a function of *R-trace*...

9. There are multiple places where the authors described regressing some model parameters (DV, GRS etc.) against BOLD signal. My understanding is that when we say "regressing A against B", we treat A as a dependent variable and B as an independent variable. So it seems the phrasing should be regressing BOLS signals against these regressors instead of the other way.

Thank you for spotting this. We have rectified this in our revision and changed the wording in main text, methods, and supplements.

10. The bottom two lines in 14: I think what the authors wanted to write may be that DRN and dACC showed earlier outcome encoding compared to Ia (not Ia or dACC)

Absolutely. We are sorry we did not spot this. Thanks for making us aware of it. We changed the sentence accordingly:

Both DRN and dACC showed significantly earlier outcome encoding compared to left and right Ia (Supplementary Figure 4f).

11. Is the posterior density of w_f in Fig 3a over all sessions (i.e., based on the final μ and σ)? It may be worth explaining it somewhere.

Not quite. It is the posterior density of w_f for all sessions of the same experiment. We have three experiments overall, the FMRI one, and two behavioural ones. This plot is to show that the weight of w_f is

similarly positive for all three experiments. We clarified this in the main text, emphasizing again that we derive group-level point estimates of μ and σ from the model:

However, the direction of the influence of GRS critically depends on w_R . In principle, the model allows GRS to have a negative, contrasting effect ($w_R < 0$), no effect at all ($w_R = 0$), or a positive effect ($w_R > 0$). We therefore examined the fitted parameter values of w_R , **i.e. the final group-level point-estimates of μ and σ associated with w_R after fitting the full model separately for each experiment.** We found that the posterior density of the fitted w_R was overwhelmingly positive for all three experiments (Fig.3a).

Thanks again for reading the manuscript so carefully and spotting inconsistencies we did not notice. We really appreciate it.

Reviewer #3 (Remarks to the Author):

and reversal paradigm to show that macaques not only use choice-outcome associations to guide decision making, but also uses the recent overall reward state of the environment (Global reward state; GRS) to do alter those choice-outcome valuations. When GRS is high (lots of rewards around) successful choices are more likely to be repeated, but when GRS is low, loss-associated choices are less likely to be repeated. Modelling of this process suggests that this is due the GRS biasing the prediction errors associated with the task feedback. MRI scanning of task performance and integration with this model indicates important roles for the anterior insula in integrating new outcomes with the overall GRS, and for the dorsal raphe nucleus in the relative weighting of outcomes.

In general I think it is a very nice study that is well written and may also provide explanations for other effects (as the authors consider in their discussion). My comments are therefore relatively minor:

Many thanks for the positive comments. We are glad the reviewer finds our study of interest.

1) How is the GRS defined for the anova analysis? Is it the last 5 trials? If so this needs to be explicit in the text as I only eventually found it in a figure legend which suggests that the "last 5 trials" were used in general for the GLMs. Why was 5 trials chosen rather than say 3 or 8?

We realise that the behavioral GLM analysis looking at the effect of the GRS is very complex and that the history length that is investigated is critical for the analysis. We have dedicated a supplementary figure to the explanation of the regressor construction (in response to question 3 of the reviewer) and to a discussion of the length of the time window over which the previous reward history is considered. We have added an analysis that varies the time window of the GRS effect between 2 and 9 trials that shows that our effect of interest, the effect of the global reward state, is stable across all of these time windows. In addition, we have added the Matlab code that was used to run the behavioral GLM to the online repository together with the RL code. This will enable anyone who is interested to track in detail how we construct the regressors for the GLM from the data.

We have clarified the length of the GRS timewindow in the main text, added very explicit references to the supplementary figure explaining the regressor construction:

Main text:

....We tested whether the stay/switch decision was predicted, first, by the conjunctive choice-reward history of C (CxR-history), second, by the (reward-unlinked) choice history of C (C-history) and, third, by the (choice-unlinked) reward history (R-history, **going back 6 trials excluding the current one**). **Note that the regressor construction is explained in detail in Supplementary Fig.1a....**

...In turn, they switch away from a choice - even if that specific choice has been rewarded lately – more often if it is encountered when in a low GRS. **This effect was stable even when varying the length of the reward history considered (Supplementary Fig.1c)...**

This supplementary figure explain regressor construction, GLM details, and invariance of the GRS effect over different time windows (see panel 1c for details on the latter):

Supplementary Figure 1. Behavioral choice GLM. **(a)** Details of regressor construction relating to Figure 1e. Left: for each trial, two out of three options were offered (A, B, and C). Black circles indicate choices and (crossed through) drops on the left indicate (no) rewards. From this information, three sets of regressors were constructed. We used them to analyze for each trial t whether the animal would subsequently stay with the same choice made in this trial or switch. The three sets of regressors are illustrated on the right side: Firstly, to capture conjunctive reward effects (green, CxR-history), we entered regressors that indicated whether choices of C on trial t and also on the previous trials were rewarded or not (CxR(t), CxR($t-1$), ...). Note that “ t ” here refers to actual occurrences of choices of C and not necessarily to consecutive trials. We only selected trials on which C was actually chosen for this regressor as only those are informative about the conjunctive choice-reward history of C. In other words, CxR-history regressors were coded for past trials in which C was chosen and set to 1/0 for rewarded/unrewarded outcomes. Reward obtained in conjunction with C should increase the probability of a stay choice. However, the stay/switch choice should also depend, inversely, on the conjunctive reward history of the alternative option that was offered as an alternative to C. To account for these effects, we created analogous regressors for the alternative option (see green weights on the right of panel b). Then, for simplicity, we aggregated the beta weights from both sets of regressors – those relating to C and those relating to the alternative option with which it was subsequently paired (see panel b for details on the aggregation). Secondly, we built regressors to account for the recent choice history of C (C($t-1$), etc.). For this, we considered whether in past trials in which C was among the offered options, C was chosen or not, irrespective of any reward receipt. Again, “ t ” refers only to a subset of trials (the ones in which C was among the offered options), but a different subset compared to the CxR regressors. As the reward history for C was captured by the CxR regressors, C-history reflects the strength of the choice trace for C irrespective of whether C was recently rewarded or not. C-history regressors were coded for past trials in which C was offered

and set to 1/0 for choosing/not choosing C on that trial. Again, we constructed analogous regressors for the alternative option that would be presented with C on the next trial and aggregated the results. Finally, we took the simple average reward on the six trials before t as an index of the overall current levels of reward (R-history). This time, “t” refers to the actual six most recent trials as these are most relevant for estimating reward history regardless of whether C is chosen or rewarded. However, we note that the results do not change with changes in history length. **(b)** Complete set of beta weights for the GLM analysis in Fig.1f. As highlighted in panel a and in the main text, we also accounted for reward related and choice repetition related effects of the upcoming alternative option (effects on the right-hand side). For Fig.1f we sign-reversed the beta weights relating to the upcoming alternative option and averaged them with CxR-history and C-history of the current choice. We averaged the beta weights of CxR-history(t-1) of the current choice (second bar in the panel) with the beta weights from CxR-history(t-1) of the upcoming alternative (13th bar in the panel), the same for t-2 and so on. Following the same approach, we averaged the beta weights of C-history(t-1) of the current choice and the (again sign reversed) C-history(t-1) of the upcoming alternative choice, and so on for t-2, t-3, t-4 and t-5. **(c)** The effect of R-history on choice is stable across a broad window of reward history length. In our main GLM, R-history is calculated as the arithmetic mean of rewards occurring during the last six trials (see explanation in the legend of panel a). We repeated our main GLM and varied the length of this reward history between two and nine trials. The figure displays the significant effects of R-history in all of those GLMs. The blue circle indicates the R-history effect from the main GLM. The relatively weakest – but still significant – R-history effect was apparent when aggregating over nine trials ($t_{11} = 3.840$; $p = 0.003$). This might reflect the timescale over which the animals aggregate the general rewards state. However, it might also partly reflect the difficulties of estimating regression weights with a very large regressor set (the corresponding GLM contains 34 regressors). Note that varying the timescale of R-history makes it necessary to simultaneously adjust the timescale of the other relevant effects in the GLM. This will ensure that past choice-reward contingencies as well as choice history alone have a similar chance to account for the observed choices as R-history. For this reason, we concurrently varied the history length over which CxR-history and C-history were calculated. For example, when reducing the history length of R-history from six to five trials, we also reduced the history length of CxR-history and C-history by one trial (for both the current choice and the upcoming alternative choice; see description in panel a) and so on. Panels b,c concatenate sessions per monkey per experiment resulting in three data points per individual; error bars are SEM across monkey data sets; $n = 12$; Source data are provided as Source Data file. Symbols indicate monkey identity in panels b,c; “MK” abbreviates “monkey”.

2) Page 30 deals with the GLM applied to simulated choices, and I'm not clear if this is identical to the one used for real choices (p6)?

It is in fact a different one. We have clarified this in the text as follows.

Main text:

We varied w_R (for illustration even beyond the bounds used above) and then analyzed the simulated choices with a GLM akin to that used in previous work (but different to the GLM in our Fig.1) to investigate spread of reward^{11,13}

Methods:

Then, we concatenated sessions from the same monkey within each experiment (just as had been done in the behavioral analysis in Figure 1) and applied a behavioral GLM to the simulated choices. This behavioral GLM is explained below and is different from the behavioral GLM in Fig.1.

3) I have missed the definitions of how CxR-history, C-history, and R-history (as per p6) were calculated?

We have now made a very explicit reference to the regressor construction which is in Supplementary Fig.1. We hope that this is helpful. See also our response to question 1. We would be happy to move this part of the manuscript from the supplements to the main text if this is in line with the editorial requirements for this manuscript regarding word count.

...We tested whether the stay/switch decision was predicted, first, by the conjunctive choice-reward history of C (CxR-history), second, by the (reward-unlinked) choice history of C (C-history) and, third, by the (choice-unlinked) reward history (R-history, **going back 6 trials excluding the current one**).

Note that the regressor construction is explained in detail in Supplementary Fig.1a....

4) Why was the influence of the R trace only investigated for prediction errors? For example, it could also have been used to influence the softmax temperature of reward-based decision-making?

That's a very interesting question. When we first conceptualised our model we based it on our behavioural findings as well as on previous models that assume an influence of the average reward rate on prediction error. We realise now that there are indeed other possibilities regarding how the reward trace could impact choice and one is via the temperature parameter of the softmax function as highlighted by the reviewer. Reviewer 1 raised a similar point by saying that the reward trace could impact choices via the CS-trace or the CL-trace, i.e. the two types of choice memory traces in our model. All of these suggestions have in common that they assume an impact of the R-trace on choice, but one that is not mediated by the PEs, but instead via another mechanism. We think that these alternative models are somewhat more implausible than our model (particularly the ones assuming an impact of the R-trace via choice traces). However, we have tested this empirically by running 8 new control models. We have added all of these control models to the online repository together with the other Matlab RL code so that it is possible for everyone to track the implementation of the models. First we explain the implementation of the models and then show the changes this has led to in the manuscript.

To start with, in our current model comparison, we consider models with a choice location trace (CL-trace), a choice stimulus trace (CS-trace) as well as a reward trace (R-trace) that modulates PEs. We consider all possible permutations of these mechanisms and find that a model including all three memory traces (RL-cl+cs+rt) fits the observed behavior best. We refer to this model also as the "full model". The control models assume, just as the full model, an effect of R-trace, but one that is conveyed not via a modulation of PEs but instead via affecting non-value-related mechanisms such as the CL-trace or CS-trace weights or the beta parameter of the softmax function. Such a modulation of choice memories could in principle generate behavior similar to the one we had observed – that animals increasingly stay with rewarded choices in a high GRS, but preferably switch away from options in low GRS environments. Such modes of action of R-trace are possible, albeit somewhat unlikely in our view. This is because, whereas the effect of the reward trace as part of the PE calculation remains within the realm of value in the full model, such an alternative account would imply that non-contingent reward memories can only be pressed into action in interaction with non-contingent choice memories. While this is not impossible, it would require us to always first assume that decision making is influenced by non-contingent choice memories and that additional effects of non-contingent reward traces are always secondary to and mediated by these choice memories. Nevertheless, we constructed such control models and show empirically that they did not fit the data as well as the full model.

We constructed 8 control models in this category. We considered that R-trace modulates the CL-trace, CS-trace, both CL- and CS-trace, or the softmax temperature beta. These are four possible alternative mechanisms and for each we constructed two versions of control models, one multiplicative and one additive. In the multiplicative version, we multiplied the respective weight with R-trace. This meant that e.g. the tendency to repeat the same stimulus choice is higher with a high R-trace and lower with a low R-trace. These models do in fact have the advantage compared to the full model that they use one less free parameter (w_R). However, note also that in these models, the impact of choice memory traces is entirely dominated by R-trace, meaning that if R-trace is zero, any tendency to repeat choices is abolished. This might not reflect the way in which R-trace modulates choice traces; instead, it might be more plausible to assume a constant effect of choice repetition that is then to some degree upscaled or downscaled by R-trace. This is why we constructed the additive control models. These make use of a free parameter that scales the additive impact of R-trace in addition to the other weight. We tested all of these models against the full model and, as mentioned before, none of them fits better than our full model.

We present the additional model comparison in the supplements as follows. The models discussed just above form the first category of control models: (see figure label “r-Trace effect not mediated by PE”)

Supplements:

Supplementary Figure 4. (a-i) ... (j,k) Supplementary control model analyses showing BICint and exceedance probabilities for the full model as well as all supplementary models. The model comparison includes all supplementary models as explained in the Methods section. There are two broad categories of control models: models that assume an influence of R-trace, but one that is not mediated by PEs but by different mechanisms; and a second category that assumes no R-trace, but instead that value updates have different dynamics. The full model RL+cl+cs+rt is the winning model according to both BICint and exceedance probability (exceedance probability = 0.959). The red line indicates an exceedance probability of 0.95. Source data are provided as Source Data file.

Methods explaining details of the modelling:

Supplementary control models. In our main model comparison, which follows on from our behavioral analyses, we formalize the effects of non-contingent choice and reward memories on decision making in an RL framework. We consider models with a choice location trace (CL-trace), a choice stimulus trace (CS-trace), and a reward trace (R-trace) that modulates PEs. We consider all possible permutations of these mechanisms and find that a model including all three memory traces (RL+cl+cs+rt) fits the observed behavior best. We refer to this model also as the “full model”. However, in a supplementary model comparison we also consider two additional categories of control models.

The first category assumes, just as the full model, an effect of R-trace, but one that is conveyed not via a modulation of PEs but instead via affecting non-value-related mechanisms such as the CL-trace or CS-trace, or the β parameter of the softmax function (equation 4). Such a modulation of choice memories could in principle generate behavior similar to the one we had observed – that animals increasingly stay with rewarded choices in a high GRS, but preferably switch away from options in low GRS environments. Similar effects are possible if R-trace modulated the β parameter because it might increase or decrease random exploration at times of low or high GRS. Such modes of action of R-trace are possible, albeit somewhat unlikely in relation to the choice traces. This is because, when reward trace exerts its impact via the PE calculation its effects remain within the realm of value in the full model. However, the alternative account would imply that non-contingent reward memories can only be pressed into action in interaction with non-contingent choice memories. While this is not impossible, it would require us to always first assume that decision making is influenced by non-contingent choice memories and that additional effects of non-contingent reward

traces are always secondary to and mediated by these choice memories. Nevertheless, we tested a comprehensive set of alternative models that incorporated such R-trace mechanisms. These alternative models mimicked the full model but eliminated the effect of R-trace from the PE calculation (equation 14) and thereby also removed the free parameter w_R that determined the weight of R-trace on the PE. Instead, in the alternative models, R-trace modulated the weights of CL-trace, CS-trace or both. We denote the new models by the interaction type of R-trace with the respective choice trace, e.g. “RT × CL”, “RT × CS” or “RT × CL&CS”. We implemented the new models by changing equation 13 in the following ways (note that we explicitly write down the multiplication symbol as × for clarity):

$$23) DV_{RT \times CL,t} = DV_{RLsimple,t} + w_{CL} \times R\text{-trace}_t \times CL\text{-bonus}_t + w_{CS}CS\text{-bonus}_t$$

$$24) DV_{RT \times CS,t} = DV_{RLsimple,t} + w_{CL}CL\text{-bonus}_t + w_{CS} \times R\text{-trace}_t \times CS\text{-bonus}_t$$

$$25) DV_{RT \times CL\&CS,t} = DV_{RLsimple,t} + w_{CL} \times R\text{-trace}_t \times CL\text{-bonus}_t + w_{CS} \times R\text{-trace}_t \times CS\text{-bonus}_t$$

These models mediate the impact of R-trace on the decision variable via the choice memory traces. These models do in fact have the advantage compared to the full model that they use one less free parameter (w_R). However, note also that in these models, the impact of choice memory traces is entirely dominated by R-trace, meaning that if R-trace is zero, any tendency to repeat choices is abolished. This might not reflect the way in which R-trace modulates choice traces; instead, it might be more plausible to assume a constant effect of choice repetition which is then to some degree upscaled or downscaled by R-trace. For this reason, we constructed additional similar control models that assume an additive effect of R-trace on the choice memory trace weight via an additional free parameter. We denote these models with “RT × CL 2”, “RT × CS 2” or “RT × CL&CS 2”:

$$26) DV_{RT \times CL\ 2,t} = DV_{RLsimple,t} + (w_{CL} + w_{RT \times CL} \times R\text{-trace}_t) \times CL\text{-bonus}_t + w_{CS}CS\text{-bonus}_t$$

$$27) DV_{RT \times CS\ 2,t} = DV_{RLsimple,t} + w_{CL}CL\text{-bonus}_t + (w_{CS} + w_{RT \times CS} \times R\text{-trace}_t) \times CS\text{-bonus}_t$$

$$28) DV_{RT \times CL\&CS\ 2,t} = DV_{RLsimple,t} + (w_{CL} + w_{RT \times CL} \times R\text{-trace}_t) \times CL\text{-bonus}_t + (w_{CS} + w_{RT \times CS} \times R\text{-trace}_t) \times CS\text{-bonus}_t$$

In close analogy to these control models assessing a mediation of R-trace effects via choice memory traces in a multiplicative or additive fashion, we also considered whether R-trace acts by modulating the inverse temperature parameter from the softmax function (equation 4). We constructed two new models, again without and influence of R-trace on the PE calculation. Instead, R-trace modulated the β parameter of these models. The new β parameters for these models were constructed as:

$$29) \beta_{RT \times \beta,t} = \beta \times R\text{-trace}_t$$

$$30) \beta_{RT \times \beta\ 2,t} = \beta + w_{RT \times \beta} \times R\text{-trace}_t$$

All of the above models belong to the first category of supplementary control models we examined. They all have in common that they assume an effect of R-trace on choice. However, they all assume that this effect is mediated via different routes compared to the one we use in the full model, i.e. instead of influencing value update in the PE, they assume that R-trace acts on non-value-related mechanisms of the model.

By contrast, the second category of control models that we examined does not require a reward trace at all. As explained in the main text, the full model effectively implements asymmetric (i.e. different for positive and negative outcomes) and dynamic (depending on whether R-trace is low or high) value updates. The first control model from this category therefore examines whether a model with no reward trace, but instead asymmetric learning rates^{48,70} fits the data better than the full

model. This model (“AsyAlpha”) makes use of the architecture of the full model without R-trace. This also removed the weight parameter, w_R , of R-trace, as well as the R-trace specific learning rate α_R . Instead, the AsyAlpha model uses separate learning rates for rewarded and for unrewarded trials. The final control model is also similarly based on the architecture of the full model without R-trace but assumes a dynamic learning rate (“DynAlpha”). It is based on the widely held idea that learning rates should increase when surprising outcomes are encountered, whereas it should decrease if outcomes conform to expectations. One RL implementation of this idea tracks the slope of absolute prediction errors to upregulate or downregulate the learning rate^{29,71}. Note that this model is not, as the others, nested within more basic models. It uses the free parameter α_R as the initial learning rate on the first trial, but also to control the smoothing when tracking the absolute prediction errors (PEmag):

$$31) PEmag_t = (1 - \alpha)PEmag_{t-1} + \alpha|PE_t|$$

From this, the normalized slope of absolute prediction errors (m) is calculated:

$$32) m_t = \frac{PEmag_t - PEmag_{t-1}}{(PEmag_t + PEmag_{t-1})/2}$$

The slope is then used to generate a link function using a free parameter γ :

$$33) fm_t = \text{sign}(m_t) \left(1 - e^{-\left(\frac{m_t}{\gamma}\right)^2}\right)$$

And this link function is then used to calculate the learning rate (LR) for the current trial:

$$34) LR_t = \begin{cases} LR_{t-1} + fm_t \times (1 - LR_{t-1}) & \text{if } m_t \geq 0 \\ LR_{t-1} + fm_t \times LR_{t-1} & \text{if } m_t < 0 \end{cases}$$

Effective learning rate calculation. As noted in the main text, the effect of R-trace in the full model RL+cl+cs+rt is that it impacts the PE calculation. This means that it increases or decreases the PE based on the current GRS, which will increase or decrease the change in Q-value. Such a mechanism can produce asymmetric and dynamic changes in value updates without explicitly changing the learning rate. Our model predicts that value updates from the same outcome event are higher for rewarded trials when R-trace is high compared to low and vice versa for unrewarded trials. We specifically tested this hypothesis by calculating the *effective learning rates* from our full model. We did this by running an additional model analysis. For this analysis, we fitted a new model that did not contain an explicit R-trace. In addition, we fixed the free parameters for each session to the session-specific parameter values from the full model. The model comprised four free parameters, all of which were learning rates, but different ones for rewarded and unrewarded trials, and ones with a high (i.e. ≥ 0.5) or low (i.e. < 0.5) R-trace. Note that the R-trace information was session-specific and imported from the full model; it was not calculated or fitted in effAlpha itself. In other words, what this model did was to assume that different effective learning rates were used for positive and negative outcomes depending on high or low R-trace. It keeps all other features identical to the full model by fixing the remaining parameters to the ones from the full model. But instead of using R-trace in the PE calculation, it examines whether the effective learning rates indeed differ in the manner expected by our full model. Note that in order to test the learning rates from effAlpha for significant differences, we fitted the model session-wise via maximum likelihood estimation, because the hierarchical fit decreases the variance of individual parameters which possibly biases comparisons of free parameters.

We thank the reviewer again for the positive and helpful comments and hope that we have addressed their concerns.

Reviewers' Comments:

Reviewer #1:

Remarks to the Author:

The authors have done an excellent job responding to reviews. I really appreciate the hard work and thoughtfulness they put into the revision. This is a great paper.

Reviewer #2:

Remarks to the Author:

The authors have significantly revised the manuscripts in response to the reviews and, in my opinion, have strengthened the paper.

All my previous comments have been addressed successfully.

Mingbo Cai

Reviewer #3:

Remarks to the Author:

I think the reviewers have done an excellent job at addressing the questions raised by the reviewers. They have very thoroughly and carefully worked through each point and incorporated them into the manuscript and supplemental as appropriate.